# Constructing a synthetic pathway for acetyl-coenzyme A from one-carbon through enzyme design

Xiaoyun Lu[1], Yuwan Liu[1], Yiqun Yang[1,2], Shanshan Wang[3], Qian Wang[1,2], Xiya Wang[4], Zhihui Yan[1], Jian Cheng[1], Cui Liu[1], Xue Yang[1,2], Hao Luo[1,2], Sheng Yang[1], Junran Gou[1], Luzhen Ye[1], Lina Lu[1], Zhidan Zhang[1], Yu Guo[2,3,5], Yan Nie [3], Jianping Lin[1], Sheng Li[3], Chaoguang Tian[1], Tao Cai [1], Bingzhao Zhuo[6], Hongwu Ma [1], Wen Wang [6], Yanhe Ma[1], Yongjun Liu[4], Yin Li[7] & Huifeng Jiang [1]

Acetyl-CoA is a fundamental metabolite for all life on Earth, and is also a key starting point for the biosynthesis of a variety of industrial chemicals and natural products. Here we design and construct a Synthetic Acetyl-CoA (SACA) pathway by repurposing glycolaldehyde synthase and acetyl-phosphate synthase. First, we design and engineer glycolaldehyde synthase to improve catalytic activity more than 70-fold, to condense two molecules of formaldehyde into one glycolaldehyde. Second, we repurpose a phosphoketolase to convert glycolaldehyde into acetyl-phosphate. We demonstrated the feasibility of the SACA pathway in vitro, achieving a carbon yield ~50%, and confirmed the SACA pathway by $^{13}$C-labeled metabolites. Finally, the SACA pathway was verified by cell growth using glycolaldehyde, formaldehyde and methanol as supplemental carbon source. The SACA pathway is proved to be the shortest, ATP-independent, carbon-conserving and oxygen-insensitive pathway for acetyl-CoA biosynthesis, opening possibilities for producing acetyl-CoA-derived chemicals from one-carbon resources in the future.

[1] Key Laboratory of Systems Microbial Biotechnology, Tianjin Institute of Industrial Biotechnology, Chinese Academy of Sciences, 300308 Tianjin, China. [2] University of Chinese Academy of Sciences, 100049 Beijing, China. [3] Shanghai Institute for Advanced Immunochemical Studies, ShanghaiTech University, 201210 Shanghai, China. [4] School of Chemistry and Chemical Engineering, Shandong University, 250100 Jinan, Shandong, China. [5] School of Life Science and Technology, ShanghaiTech University, 201210 Shanghai, China. [6] Center for Ecological and Environmental Sciences, Northwestern Polytechnical University, 710072 Xi'an, China. [7] Key Laboratory of Microbial Physiological and Metabolic Engineering, State Key Laboratory of Microbial Resources, Institute of Microbiology, Chinese Academy of Sciences, 100101 Beijing, China. These authors contributed equally: Xiaoyun Lu, Yuwan Liu, Yiqun Yang, Shanshan Wang. These authors jointly supervised this work: Huifeng Jiang, Yin Li, Yongjun Liu. Correspondence and requests for materials should be addressed to Y.L. (email: yongjunliu_1@sdu.edu.cn) or to Y.L. (email: yli@im.ac.cn) or to H.J. (email: jiang_hf@tib.cas.cn)

Acetyl-CoA is a hub metabolite in central metabolic pathways for all living life, which harnesses the catabolism and anabolism of almost all fundamental nutrients, such as sugar[1], fat[2,3], and protein[4] in cells. As a sole donor of acetyl group, acetyl-CoA also provides the C2 group for numerous biochemical synthesis of industrial chemicals[5–7] and natural compounds[8,9]. Nature has evolved several pathways to synthesize acetyl-CoA, such as glycolysis pathway through decarboxylation of pyruvate by pyruvate dehydrogenase[10], phosphoketolase (PK) pathway through degrading fructose-6-phosphate (F6P) or xylulose-5-phosphate (X5P) by PKs[11], serine cycle pathway through dividing malyl-CoA by malyl-CoA lyase[12,13], Wood–Ljungdahl (WL) pathway through condensation of one-carbons by acetyl-CoA synthase (ACS)[14]. Although many efforts have been done[15–19], engineering the natural biosynthetic pathways of acetyl-CoA restricted by many inherent defects such as carbon loss, ATP dependent and extremely anaerobic condition.

Since one-carbon assimilation greatly reduce industrial production cost and also would alleviate the pressure of resource supplement for bio-manufacturing[20,21], recent years, several artificial pathways have been constructed to produce acetyl-CoA from one-carbon resources. For example, the formolase (FLS) pathway could condense three molecules of formaldehyde (FALD) to one molecule of dihydroxyacetone by the designed formolase and then produce acetyl-CoA via glycolysis pathway[22]. Non-oxidative cyclic glycolysis (NOG) pathway achieved the biosynthesis of acetyl-CoA without carbon loss by rewiring the known carbon rearrangement pathway[23]. Methanol condensation cycle (MCC) pathway produced acetyl-CoA by combining the ribulose monophosphate pathway (RuMP) and NOG pathway for carbon conservation and ATP-independent one-carbon assimilation[24]. By simplifying the enzymatic steps, the modified serine cycle also assimilate methanol (or formate) and bicarbonate into acetyl-CoA[13]. However, these designed pathways from one-carbon to acetyl-CoA largely overlapped with the known metabolic network, it is difficult to assign metabolic flux among the designed pathways and the original metabolic pathways.

In this study, we de novo designed a synthetic pathway from formaldehyde to acetyl-CoA without overlapping with the known metabolic network. By combining protein design and pathway construction in vitro and in vivo using *Escherichia. coli* as host, we realized the biosynthesis of acetyl-CoA from formaldehyde. It will be possible to plug this synthetic pathway into the central metabolic network to provide acetyl-CoA from one-carbon resources for bulk chemicals biosynthesis and even life living.

## Results

**Design of the synthetic acetyl-CoA pathway.** The simplest way to synthesize organic carbon from one-carbon is to construct it one by one. In order to build an artificial acetyl-CoA pathway from one carbon, we proposed the Synthetic Acetyl-CoA (SACA) pathway, where two molecules of formaldehyde would be transferred into one molecule of acetyl-CoA through only three steps (Fig. 1a). Firstly, formaldehyde would be condensed into glycolaldehyde (GALD) by the glycolaldehyde synthase (GALS). And then glycolaldehyde would be converted into acetyl-phosphate (AcP) by the acetyl-phosphate synthase (ACPS) using inorganic phosphate. Finally, AcP would be used to produce acetyl-CoA by the known enzyme phosphate acetyltransferase (PTA)[25]. Meanwhile, formaldehyde can be obtained by reducing carbon dioxide and formate[26] or oxidizing methane and methanol[27]. Thus, we could realize the biosynthesis of acetyl-CoA from formaldehyde and even other one-carbon resources.

The thermodynamics can reflect whether a pathway could be effectively carried out in vivo or in vitro. We calculated the thermodynamic chemical driving force of the designed SACA pathway. The overall reaction from formaldehyde to acetyl-CoA is highly thermodynamically favorable, where the total Gibbs energy change ($\Delta_r G'^m$) of the whole reaction is about $-96.7$ kJ

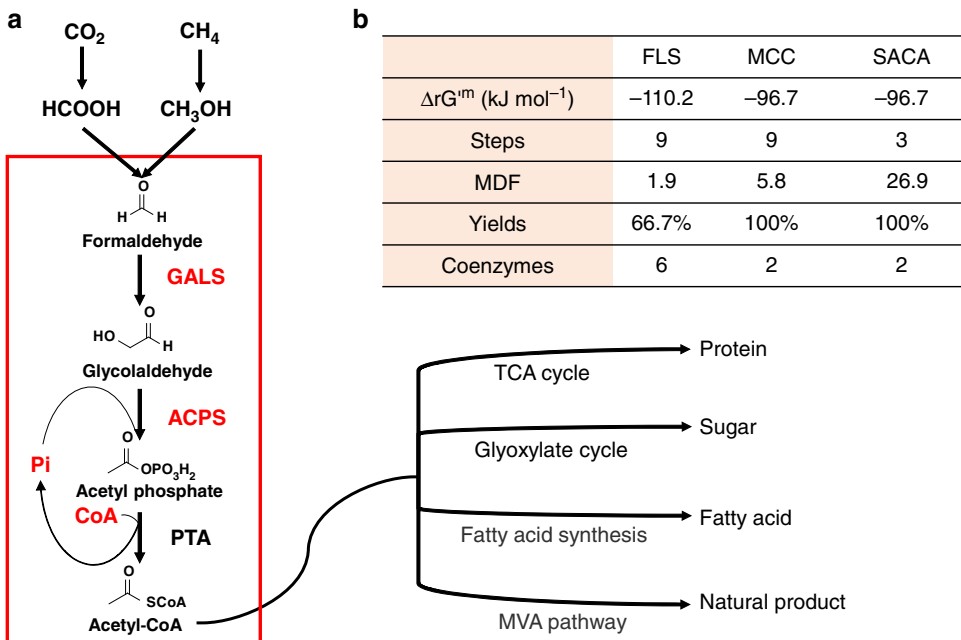

| | FLS | MCC | SACA |
|---|---|---|---|
| $\Delta_r G'^m$ (kJ mol$^{-1}$) | −110.2 | −96.7 | −96.7 |
| Steps | 9 | 9 | 3 |
| MDF | 1.9 | 5.8 | 26.9 |
| Yields | 66.7% | 100% | 100% |
| Coenzymes | 6 | 2 | 2 |

**Fig. 1** Description and computational analysis of the SACA pathway. **a** The SACA pathway was highlighted in the red panel. The main feedstock of formaldehyde could be from methanol, formate and even methane and CO₂. The product of acetyl-CoA could be used to generate major cellular nutrients. **b** The thermodynamic data of three designed pathways for acetyl-CoA synthesis were generated by the website of eQuilibrator (http://equilibrator. weizmann.ac.il). Δ$_r$G'$^m$: the total Gibbs energy change; Steps: the number of reactions from formaldehyde to acetyl-CoA in the studied pathways; MDF: the maximum driving force; Yields: the total yield of carbon in the studied pathways; Coenzymes: the number of coenzymes is used in the studied pathways

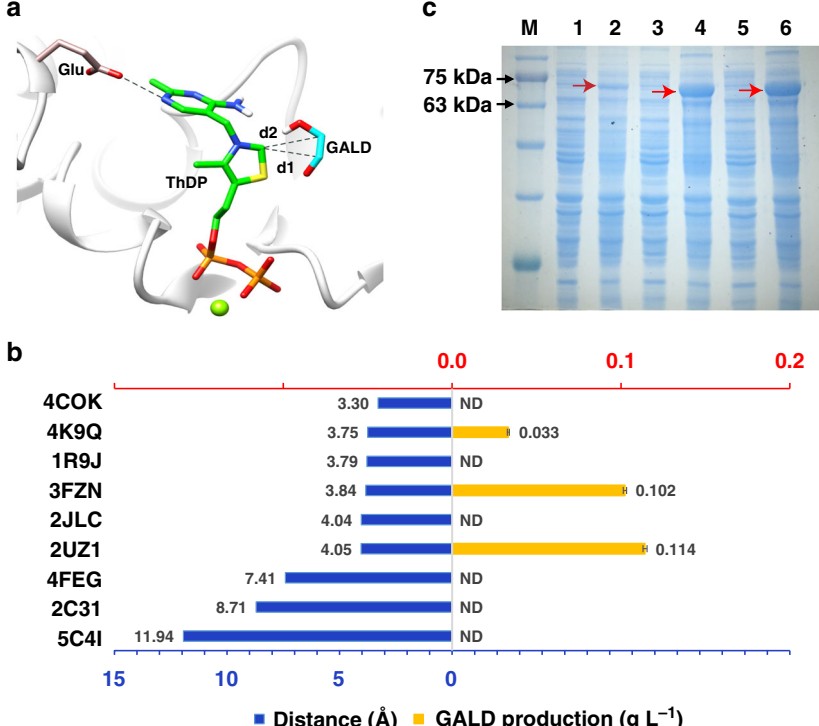

**Fig. 2** Theozyme model construction and functional identification of the glycolaldehyde synthase. **a** The theozyme model interaction between glycolaldehyde and the active centers of different ThDP-dependent enzymes. The glutamate (glu) is tan; glycolaldehyde is cyan; ThDP is green. The distances between C2 atom in ThDP and glycolaldehyde carbon atoms are represented by d1 and d2. The green dot is magnesium ion. **b** The average distances (blue) between d1 and d2 in each protein are shown on the left. The amount of product (yellow) for the tested protein are shown on the right. The reaction was carried out by adding 1 mg mL$^{-1}$ of the tested proteins and 2 g L$^{-1}$ formaldehyde. ND no detection. Error bars represent s.d. (standard deviation), $n = 3$. **c** Protein expression of three functional candidates by using 1 mL of 1 OD cells. M: protein marker; 1, 3 and 5 represent protein expression without IPTG for 2UZ1, 3FZN, and 4K9Q, respectively; 2, 4, and 6 represent protein expression under IPTG inducing for 2UZ1, 3FZN, and 4K9Q, respectively; The red arrows point to the protein bands for 2UZ1, 3FZN, and 4K9Q, respectively. Source data are provided as a Source Data file.

mol$^{-1}$ (Fig. 1b and Supplementary Table 1). The value of MDF (maximum driving force) is usually used to evaluate the thermodynamic and kinetic quality of different pathways[28]. If the MDF is sufficiently high, the pathway contains no thermodynamic bottlenecks that would hamper its operation in vivo. The SACA pathway obtained the relatively high MDF value of 26.9 kJ mol$^{-1}$, which is obviously higher than the FLS and MCC pathways (their MDF values are 1.9 and 5.8 kJ mol$^{-1}$, respectively). Therefore, the SACA pathway is thermodynamically favorable for the biosynthesis of acetyl-CoA from formaldehyde.

**Identification of a novel enzyme from C1 to C2.** The condensation of formaldehyde can be catalyzed by the N-heterocyclic carbine in chemistry[29,30]. In biology, the thiazolium ring of the cofactor thiamine diphosphate (ThDP) has similar function, which could activate one aldehyde and then form dimer with another aldehyde[31]. In order to find an enzyme to condense two molecules of formaldehyde into one molecule of glycolaldehyde, we referred to the catalytic mechanisms of ThDP-dependent enzymes and constructed a theozyme model, which includes ThDP, glycolaldehyde, and glutamic acid that provides electron for the reaction[32] (Fig. 2a). All enzymes in Protein Data Bank (PDB) were virtually screened based on the theozyme model and 37 non-redundant protein structures with ligand ThDP were achieved (Supplementary Fig. 1 and Supplementary Note). According to the catalytic mechanisms of ThDP-dependent enzymes[33–35], C2 atom in ThDP is the active center. The distance between C2 atom and the product of glycolaldehyde is critical for triggering the catalytic reaction (Fig. 2a). Thus, we analyzed the

distance between C2 atom and glycolaldehyde in each candidate protein.

Based on the average distances in each protein using molecular docking (Supplementary Data 1)[36], six candidates with short distances and clear functional annotations were defined as candidates (Fig. 2b and Supplementary Table 2). In addition, three proteins with long distances were randomly chosen as controls. The candidates and controls were expressed and purified to test their ability of producing glycolaldehyde from formaldehyde. Three out of six candidates exhibited the desired activity, while three controls did not have the function, indicating the distance between C2 atom and glycolaldehyde plays a critical role on the condensation of formaldehyde. Among three active candidates, the protein 2UZ1 (https://www.rcsb.org/structure/2UZ1) that was termed benzaldehyde lyase (BAL), was reported to preferentially generate dihydroxyacetone (DHA) in spite of the minimal yield of glycolaldehyde when the concentration of formaldehyde was lower[37]. The other two proteins 3FZN (https://www.rcsb.org/structure/3FZN) and 4K9Q (https://www.rcsb.org/structure/4K9Q) that produce glycolaldehyde from formaldehyde were named as benzoylformate decarboxylases (BFD) in PDB database. And BFD (3FZN) from *Pseudomonas putida* was selected as candidate for subsequent modification since it has higher activity and expression level (Fig. 2c).

**Directed evolution of the glycolaldehyde synthase.** Since BAL had been engineered to produce DHA from formaldehyde[22], we proposed to detect if the corresponding mutations in BFD would also contribute to improve the enzyme activity (Supplementary

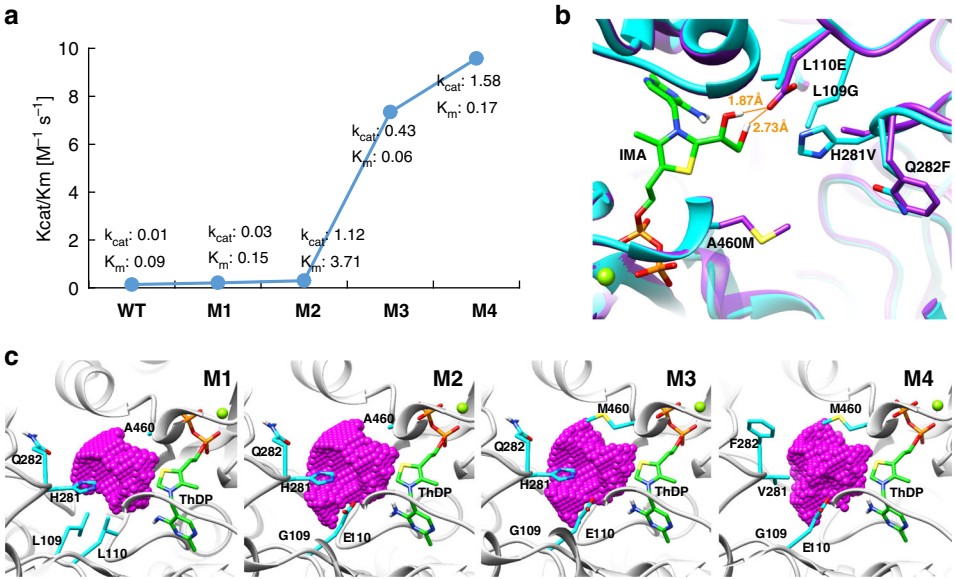

**Fig. 3** Protein engineering and mechanism analysis of the glycolaldehyde synthase. **a** Kinetic parameters of WT and mutants. WT: wild type; M1: mutations at W86R and N87T; M2: mutations at W86R, N87T, L109G, and L110E; M3: mutations at W86R, N87T, L109G, L110E and A460M; M4: mutations at W86R, N87T, L109G, L110E, A460M, H281V, and Q282F. **b** The overview of the selected five mutations in active center. IMA: intermediate analogue; the orange lines indicate the hydrogen bonds between the hydroxyl group of IMA and the mutation L110E. **c** The pocket volumes of M1, M2, M3, and M4. Pink dots represent the volumes of the binding-pockets (Supplementary Data 2), which are 131.25, 161.50, 133.38, and 171.38 Å³, respectively. The figures were rendered using UCSF Chimera software version 1.12[46]. Source data are provided as a Source Data file.

Fig. 2). By screening all mutated residues, we indeed found a highly active mutation W86R-N87T which was located in the substrate channel of BFD (Supplementary Fig. 3). Thus this mutation (W86R-N87T) was introduced into BFD and the variant was marked as M1. In order to further improve the catalytic activity of BFD, we proposed to screen all residues around the active center of BFD, where 25 positions within 8 Å distance from the active center were selected to do single-point saturation mutagenesis (Supplementary Fig. 4). We developed a high-throughput screening approach to detect glycolaldehyde by color reaction between glycolaldehyde and diphenylamine, which was measured by spectrophotometrically monitor at 650 nm (Supplementary Fig. 5). After screening, we found that 14 out of 25 positions showed higher activities than M1 mutant (Supplementary Fig. 6). Subsequently, 14 positions were divided into 8 groups: A (N27/G402), B (N27/S236), C (N27/A460), D (F397/C398), E (L109/L110), F (H281/Q282), G (N374/S378), and H (T379/T380). N27 was used three times since the variants in this position displayed the highest activity. Using M1 as template, we introduced each group of mutations into M1 and selected the highest active mutant, which was labeled as M2. By performing three rounds of iterative combinatorial mutagenesis among these positions, we totally screened 64,512 clones and obtained a high activity mutant that contains five novel mutations around the active center. Finally, the variant with 7 residue mutations was named as glycolaldehyde synthase (GALS). The $k_{cat}$ of GALS was improved about 160-fold and the final catalytic efficiency of GALS is 9.6 M⁻¹·s⁻¹, which is roughly 70-fold than the starting enzyme (Fig. 3a and Supplementary Fig. 7).

In order to figure out how these mutations improve enzyme activity, we proposed to redo crystallization of GALS. The active pocket locates at the interface of homodimer of GALS[38]. The protein structure of GALS may differ from the starting protein after several rounds of mutation. Here, we crystallized the protein of GALS and retrieved the crystal structure of GALS using the structure of BFD as the search model (Supplementary Table 3). The crystal structure of GALS was analyzed (Supplementary

Fig. 8, Supplementary Fig. 9). We found that the mutation of L110E introduces two additional hydrogen bonds to the hydroxyl group of intermediate analogue (IMA), which may contribute in stabilizing transition state and cleaving C–C bond between product and cofactor ThDP (Fig. 3b). The mutation of L109G enlarged the volume of substrate binding pocket by replacing the big isobutyl group with a hydrogen group. The third mutation of A460M may reorient the substrate and then enhance the interaction between ThDP and substrate. The last two mutations H281V and Q282F expanded the pore radius of outside surface and may facilitate access for substrate or product (Fig. 3c). Comparing with M1, the volume of reaction pocket in GALS was enlarged more than 30%, which would be the main reason for the improvement of catalytic activity.

**Identification of the acetyl-phosphate synthase.** In nature, no enzyme was reported to achieve the synthesis of AcP from glycolaldehyde. Phosphoketolases (PKs) can produce AcP from fructose-6-phosphate (F6P) or xylulose-5-phosphate (X5P)[39]. According to the catalytic mechanism of PKs, it is possible that glycolaldehyde interacts with ThDP and then generates 2-α,β-dihydroxyethylidene-ThDP (DHEThDP) (Fig. 4a), which is the key intermediate of forming AcP from F6P or X5P by PKs. To confirm our hypothesis, we selected eight candidates (Fig. 4b) based on the phylogenetic tree of PKs from 111 bacteria families (Supplementary Fig. 10). After gene synthesis and protein purification (Supplementary Fig. 11), we examined the catalytic activity of all candidate proteins using glycolaldehyde as substrate. Fortunately, five out of eight PKs displayed significant catalytic activities. Only PK1, PK4, and PK8 did not show significant difference from the blank control. Thus, PK2 with the highest activity was termed as acetyl-phosphate synthase (ACPS), whose $k_{cat}/K_m$ achieves to 3.21 M⁻¹·s⁻¹ (Supplementary Fig. 12).

To reveal the mechanism of forming DHEThDP from glycolaldehyde, we proposed to perform theoretical analysis based on the previous computational model of PK[40]. All possible

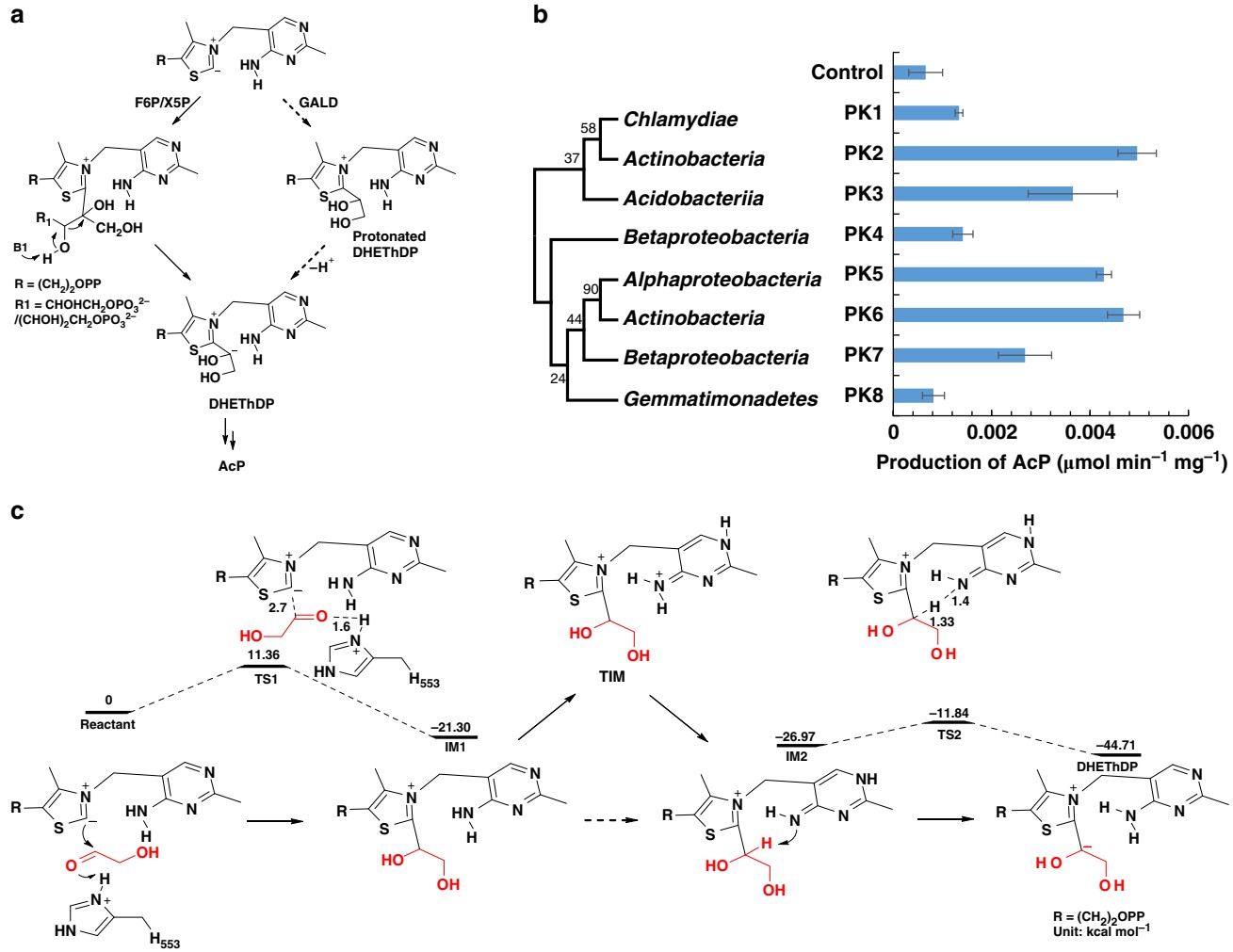

**Fig. 4** Computational analysis and functional identification of the acetyl-phosphate synthase. **a** The formation of DHEThDP from F6P/X5P and glycolaldehyde. The solid arrows represent the formation mechanism of DHEThDP in ref. [39]; the dashed arrows indicate the predicted process using glycolaldehyde as substrate. **b** The identification of ACPS. The maximum-likelihood phylogenetic tree of the selected eight PKs was constructed by MEGA[47]. The details of the selected PKs are present in the supplementary note and supplementary Fig. 10. The activity of each protein was detected by adding 0.5 mg mL$^{-1}$ enzyme and 10 mM glycolaldehyde. AcP acetyl phosphate, PK phosphoketolase, Production of AcP (μmol min$^{-1}$ mg$^{-1}$): the amount of AcP produced per mg enzyme per minute; Control: no enzyme was added in the reaction system; Error bars represent s.d. (standard deviation), $n = 3$. **c** The energy profile for DHEThDP formation process. IM1 and IM2 represent intermediate 1 and 2 during the formation process; TIM represents the tautomerized intermediate between IM1 and IM2; TS1 and TS2 represent the transition states. Source data are provided as a Source Data file.

amino acids related to the formation of DHEThDP were used to construct the computational model (Supplementary Fig. 13). Based on computational simulation, we found that the energy barrier is only 11.36 kcal mol$^{-1}$ for the formation of IM1 (intermediate 1) when glycolaldehyde was protonated by His553, which was considered as the most possible proton donor[40] (Fig. 4c). Then, IM1 tautomerizes into IM2 with the assistance of Glu479 and Glu437. IM2 is energetically more stable than IM1. When the proton of IM2 was stripped off by N4' in ThDP, the energy barrier from IM2 to the key intermediate DHEThDP is 15.13 kcal mol$^{-1}$, which is as similar as the strain energy during the formation of DHEThDP using F6P as substrate[40]. After the formation of DHEThDP, the following catalytic processes are the same as other PKs (Supplementary Fig. 14), i.e., H97 acts as proton donor of dehydration process of DHEThDP, and His142 and Gly155 accelerate the dehydration process of DHEThDP, judging from the fact that His142 and Gly155 form hydrogen bonds with the water molecule in structure of the post-dehydration intermediate (AcThDP). His64, Tyr501, and Asn549 are important for the nucleophilic

attack by Pi and they together form the binding site of Pi[39]. Site-directed mutagenesis experiments also indicated that these residues are pivotal for enzyme activity (Supplementary Fig. 15).

**Synthesis of acetyl-CoA from formaldehyde in vitro**. With the initial successful design of GALS and ACPS, we intended to construct the SACA pathway in vitro to synthesize acetyl-CoA from formaldehyde. Firstly, we measured the yield of glycolaldehyde by GALS under different concentrations of formaldehyde. The results showed that GALS can effectively catalyze the dimerization of formaldehyde, and the yield of glycolaldehyde increased with the concentration of substrate improved (Fig. 5a). For example, the yield of glycolaldehyde from formaldehyde increased from 45% at 0.5 g L$^{-1}$ to 80% at 2 g L$^{-1}$. Secondly, we examined the catalytic efficiency from glycolaldehyde to AcP, which was quantified by the content of acetic acid since AcP would be quickly degraded into acetic acid (Supplementary Fig. 16)[23]. The results showed that the yield of AcP reached to more than 80% under different concentrations of glycolaldehyde

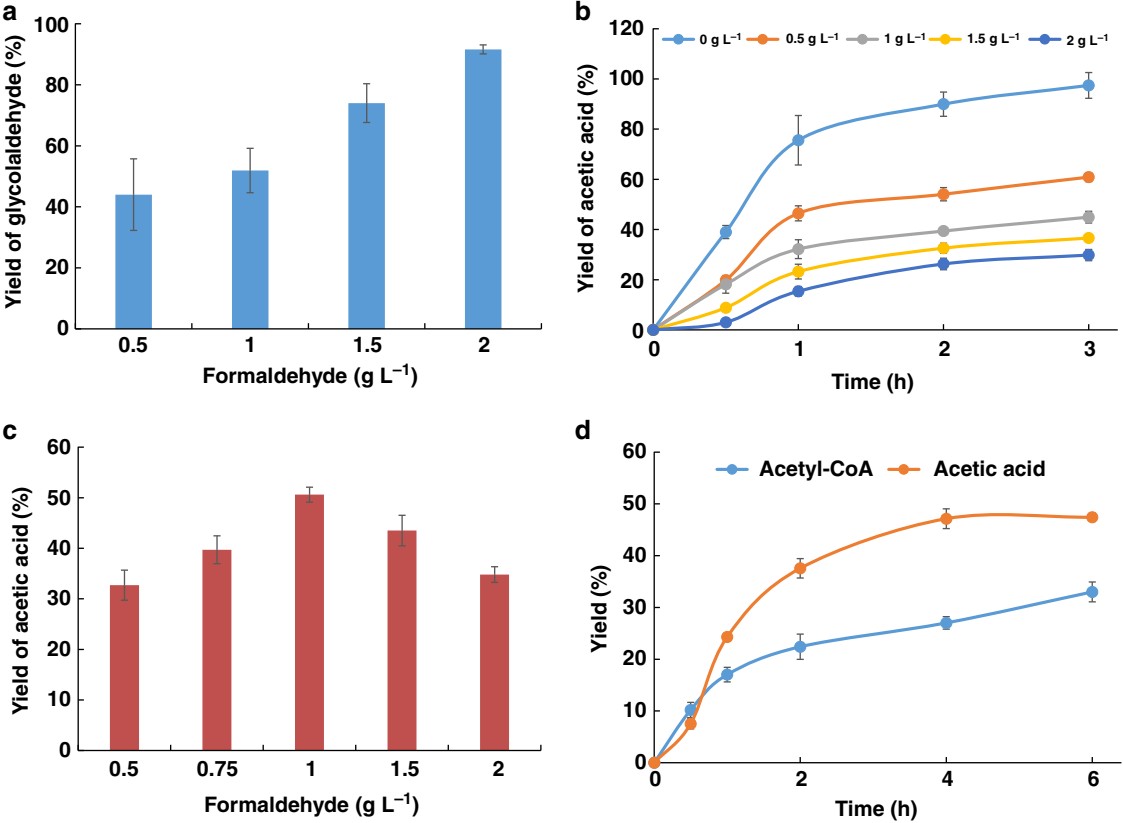

**Fig. 5** Confirming the biosynthesis of acetyl-CoA from formaldehyde by the SACA pathway in vitro. **a** The synthesis of glycolaldehyde from formaldehyde by GALS. Reactions were executed under different formaldehyde concentrations with 2 mg mL$^{-1}$ purified GALS at 37 °C for 2 h. **b** The inhibition of ACPS by formaldehyde. The yields of acetic acid were measured at different time points with reaction buffer, 2 mg mL$^{-1}$ ACPS, 0.75 g L$^{-1}$ glycolaldehyde and the gradient of formaldehyde from 0 to 2 g L$^{-1}$ which was represented by different color lines. **c** Yield of acetic acid from different formaldehyde concentrations. Reactions were carried out at 37 °C overnight with reaction buffer, 2 mg mL$^{-1}$ GALS, 2 mg mL$^{-1}$ ACPS and the gradient of formaldehyde from 0.5 to 2 g L$^{-1}$. **d** Yield of acetic acid or acetyl-CoA from 1 g L$^{-1}$ formaldehyde. The yields of acetic acid or acetyl-CoA were measured at different time points. The blue line represents the reaction system to produce acetyl-CoA (with 20 mM CoA feeding and containing 2 mg mL$^{-1}$ of the purified GALS, ACPS, and PTA, respectively); the orange line represents the reaction system to produce acetic acid (containing 2 mg mL$^{-1}$ GALS and ACPS and without CoA feeding). Samples in all assays were analyzed by HPLC. Error bars represent s.d. (standard deviation), $n = 3$. Source data are provided as a Source Data file.

via ACPS (Supplementary Fig. 17). Unfortunately, ACPS was obviously inhibited by formaldehyde (Fig. 5b). Thus, it is necessary to take a balance for the concentration of formaldehyde in resolving this conflict.

In order to optimize the concentration of formaldehyde, a gradient experiment was performed using the reaction system that contains 2 mg mL$^{-1}$ of GALS and ACPS. With increasing the concentration of formaldehyde, the yield of acetic acid in the system initially increased and then decreased (Fig. 5c). When the concentration of formaldehyde is 1 g L$^{-1}$, the yield of acetic acid reached to 50.6%. Interestingly, the final yield of acetic acid is even higher than that in the reaction system from glycolaldehyde to AcP with the same amount of formaldehyde (Fig. 5b, grayline). This would be caused by partially releasing the function of ACPS since formaldehyde was gradually consumed by GALS. Based on this system, we further added another known enzyme phosphate acetyltransferase (PTA), which would convert AcP into acetyl-CoA. Finally, the SACA pathway produced 5.5 mM acetyl-CoA at the formaldehyde concentration of 1 g L$^{-1}$. The yield of acetyl-CoA from formaldehyde was about 33% (Fig. 5d). However, the yield of acetic acid (7.8 mM) from formaldehyde (33.3 mM) reached to 50% if PTA and CoA were not supplied. The lower yield of acetyl-CoA than that of acetic acid would be caused by the instability of acetyl-CoA and AcP. Our results

indicated that it is successful for the biosynthesis of acetyl-CoA from formaldehyde by the SACA pathway in vitro.

**Validation of the SACA pathway by $^{13}$C-labeling**. After ascertaining the SACA pathway with purified enzymes in vitro, we further attempted to test the biosynthesis of acetyl-CoA and it's derivates from the SACA pathway in vivo by $^{13}$C-metabolic tracer assays (Fig. 6a). Firstly, cell lysates were used to verify the biosynthesis of acetyl-CoA by the addition of $^{13}$C-labeled formaldehyde and CoA. We detected significant increase of the double $^{13}$C-labeled acetyl-CoA than other controls (P-value < 0.001, T-test) (Fig. 6b and Supplementary Fig. 18). Furthermore, the increase of double $^{13}$C-labeled acetyl-CoA disappeared if we omitted one of genes in the SACA pathway such as ACPS and PTA (Supplementary Fig. 19). In addition, acetyl-CoA would be converged with oxaloacetate to enter the tricarboxylic acid (TCA) cycle. We also detected significantly more double $^{13}$C-labeled fumarate (P-value < 0.001, T-test) and malate (P-value < 0.001, T-test) only in the strain with the SACA pathway and $^{13}$C-labeled formaldehyde feeding (Fig. 6b and Supplementary Fig. 18). However, we did not detect significant differences in higher order mass isotopomers. It would be caused by the low amount of double $^{13}$C-labeled isotopomers in the TCA cycle.

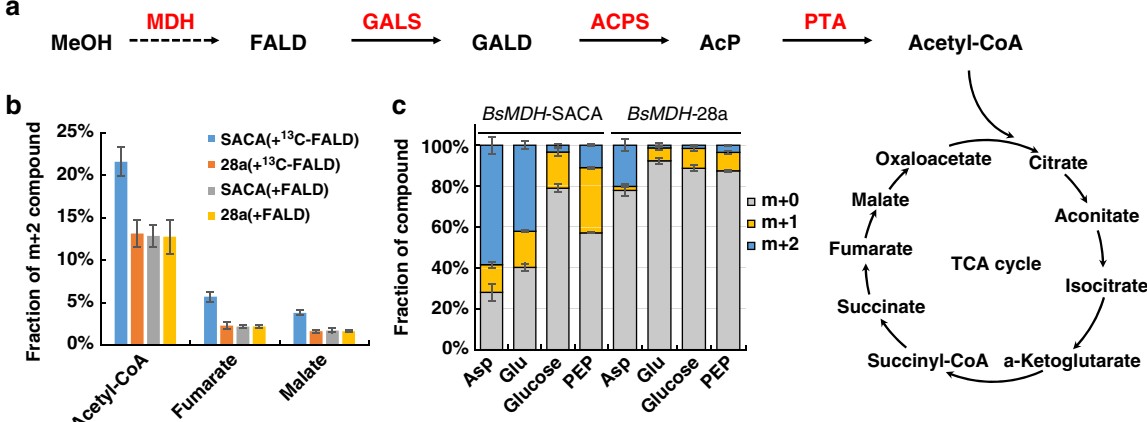

**Fig. 6** [13]C-labeled metabolic tracer analysis of the SACA pathway. **a** Schematic diagram of the tested pathway for [13]C-labeled tracer. **b** The fraction of m + 2 compounds in cellular lysates with [13]C-labeled or unlabeled formaldehyde. **c** The fraction of [13]C-labeled metabolites. Cells were induced in LB and transferred to M9 medium containing [13]C-labeled methanol. Intracellular metabolites were detected after 16 h incubation and proteinogenic amino acids were measured after 26 h incubation. The sum of the detected isotopomers was set as 100%. [13]C-FALD [13]C-labeled formaldehyde, m + 0 without [13]C labeling, m + 1 single [13]C labeling, m + 2 double [13]C labeling, FALD formaldehyde, MeOH methanol, GALD glycolaldehyde, AcP acetyl-phosphate, Asp aspartate, Glu glutamate, PEP phosphoenolpyruvate, SACA the strain contains the vector GALS-ACPS-PTA-28a, 28a the strain contains the empty vector of 28a, BsMDH-SACA the strain contains both BsMDH-pCDF and GALS-ACPS-PTA-28a vectors, BsMDH-28a: the strain contains both BsMDH-pCDF vector and the empty vector 28a. Error bars represent s.d. (standard deviation), $n = 3$. Source data are provided as a Source Data file.

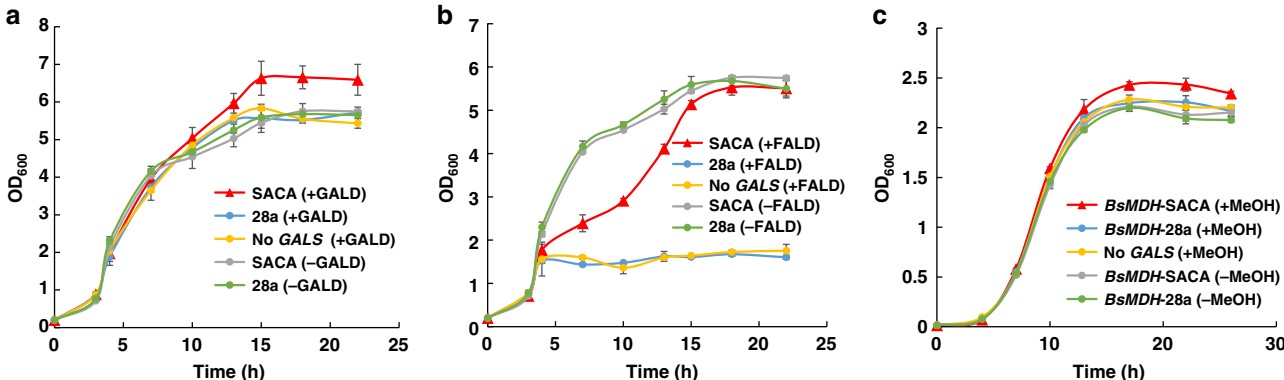

**Fig. 7** Assessing the SACA pathway by cell growth. **a** Cell growth assays using 0.4 g L$^{-1}$ glycolaldehyde as supplemental carbon source. **b** Cell growth assays using 80 mg L$^{-1}$ formaldehyde as supplemental carbon source. **c** Cell growth assays using 8 g L$^{-1}$ methanol as supplementary carbon source. Cells initially were cultured in LB medium. IPTG was added to induce protein expression and then the supplemental carbon sources were added. OD$_{600}$ was detected at different time points. SACA the strain contains the vector GALS-ACPS-PTA-28a, 28a the strain contains the empty vector of 28a, BsMDH-SACA the strain includes both BsMDH-pCDF and GALS-ACPS-PTA-28a vectors, BsMDH-28a the strain contains both BsMDH-pCDF vector and the empty vector 28a, No GALS the strain contains all genes in the pathway except GALS, + adding supplemental carbon source, − without supplemental carbon source, glycolaldehyde (GALD), formaldehyde (FALD), methanol (MeOH). Error bars represent s.d. (standard deviation), $n = 3$. Source data are provided as a Source Data file.

Subsequently, we proposed to test carbon flow within cells. Due to the toxicity of formaldehyde to cells, we introduced the methanol dehydrogenase from *Bacillus stearothermophilus* (BsMDH)[41] to maintain a continuously low concentration of formaldehyde in vivo (Fig. 6a). The cultured cells were harvested at different time points and were used to detect proteinogenic amino acids and intracellular metabolites (Fig. 6c). We found that the strain with the SACA pathway (BsMDH-SACA) contained more double [13]C-labeled aspartate and glutamate, which derived from the TCA cycle. In addition, the [13]C-labeled glucose and phosphoenolpyruvate, which might generate from double [13]C-labeled oxaloacetate through the gluconeogenesis pathway, were detected more in the strain BsMDH-SACA than those in the control strain (BsMDH-28a). Thus our results indicated that the SACA pathway is functional for producing acetyl-CoA and acetyl-CoA-derived metabolites from formaldehyde.

**Evaluation of the SACA pathway by cell growth**. In order to further evaluate the SACA pathway in vivo, we proposed to test the cell growth stepwise by feeding each intermediate in the pathway. Initially *E. coli* grew up under the rich medium and then glycolaldehyde was added as supplemental carbon source. By adding different concentrations of glycolaldehyde (Supplementary Fig. 20), we found that the engineered strain that contains the SACA pathway with more than 0.4 g L$^{-1}$ glycolaldehyde supply has remarkable higher OD$_{600}$ than those strains without glycolaldehyde or the SACA pathway (Fig. 7a and Supplementary Fig. 21). The contribution of glycolaldehyde to biomass (cellular dry weight, CDW) was $0.681 \pm 0.028$ gCDWg$^{-1}$ ($n = 3$) glycolaldehyde.

When we tested cell growth using formaldehyde as supplemental carbon source, the growth of the strain with empty vector was totally inhibited by 80 mg L$^{-1}$ of formaldehyde (Fig. 7b). The

strain containing SACA pathway was inhibited initially and then grew up normally under the same condition, which may be caused by its faster rate of formaldehyde consumption than the strain with empty vector (Supplementary Fig. 22). Unfortunately the strain containing SACA pathway did not have more biomass with supplement of formaldehyde than those without formaldehyde. These results indicated that although the SACA pathway is more efficient for removing the toxic formaldehyde, it is not enough to provide biomass.

At last, we intended to evaluate the SACA pathway by feeding methanol, which would be continuously converted into formaldehyde. We found that the strain containing both *BsMDH* and SACA pathway has higher $OD_{600}$ than those controls without methanol supply or the SACA pathway (Fig. 7c and Supplementary Fig. 23). After 26 h incubation, we found that the value of $OD_{600}$ in the strain containing both *BsMDH* and SACA pathway increased about 0.2, which is significantly higher than the strain without the SACA pathway (*P*-value = 0.005, *T*-test). By comparing with the strain without the SACA pathway, we found that the amount of biomass derived from methanol was $0.03 \pm 0.006$ gCDW g$^{-1}$ ($n = 3$) methanol in the strain that contains the SACA pathway.

## Discussion

It has been reported that more than 35% industrial chemicals would be produced by bio-manufacturing until 2030[42]. Seeking cheaper and more sustainable feedstock for bio-manufacturing is a major challenge in the field of industrial biology. Organic one-carbon resources, including methane, methanol, formaldehyde and formic acid, would be desired feedstocks, since the annual total output are extremely plentiful and would be sustainably generated from $CO_2$[26,43,44]. In this study, we successfully constructed the SACA pathway by involving two designed enzymes to convert one-carbon formaldehyde into the central metabolite acetyl-CoA, which is precursor for most of products in bio-manufacturing. The designed enzyme GALS would produce 1.8 g L$^{-1}$ glycolaldehyde within 2 h using 2 g L$^{-1}$ formaldehyde (Supplementary Fig. 24) and ACPS would generate 1.2 g L$^{-1}$ acetic acid within 2 h using 1.5 g L$^{-1}$ glycolaldehyde, which are almost close to industrial production intensity (Supplementary Fig. 17). Unfortunately, by assembling both designed enzymes together in the SACA pathway, we only achieved about 33% yields of acetyl-CoA and 50.6% yields of acetic acid from 1 g L$^{-1}$ formaldehyde since the function of ACPS was inhibited by formaldehyde (Fig. 5d). Therefore, releasing formaldehyde inhibition on ACPS would be a possible strategy improving the efficiency of the SACA pathway in the cell free system in the future.

Comparing with the considerable output for the SACA pathway in vitro, the contribution of the SACA pathway in vivo is too trivial due to the low substrate affinities for both designed enzymes ($K_m = 165$ mM for GALS and $K_m = 51$ mM for ACPS). Considering high toxicity of formaldehyde to cell, it is impossible to supply the same amount of formaldehyde in vivo. By introducing *BsMDH* that could slowly provide formaldehyde from methanol in vivo, we achieved a successful growth using the SACA pathway in vivo (Fig. 7c). Currently the SACA pathway only contributed about 3% biomass from methanol, which is obviously lower than the nature evolved pathway like RuMP pathway that contributed 24% biomass from methanol[16]. However we think that there are two strategies to improve performance of the SACA pathway in vivo. It is accessible to increase substrate affinities of both designed enzymes by protein engineering and then facilitate the application of the SACA pathway within cells. It also should be possible to consider applying other

host, like *Pichia pastoris*[45], whose peroxisomes allow to maintain formaldehyde at a higher concentration.

In summary, although it is not currently practicable in vivo, the SACA pathway exhibits the following advantages: (I) it is the shortest pathway from formaldehyde to acetyl-CoA which only contains three enzymes; (II) it is carbon-conserved and ATP-independent; (III) it is feasible under both aerobic and anaerobic conditions. Thus, continuing to improve enzyme activities in this pathway or to increase formaldehyde tolerance would enable carbon-conserving and ATP-independent conversion to produce chemicals and fuels from plentiful one-carbon resources, which could alleviate the pressure of the resource supply for bio-manufacturing and also greatly reduce production costs. Furthermore, combining with the electrochemical conversion from $CO_2$ to formic acid[43] and biotransformation from formic acid to formaldehyde[26], the SACA pathway would also be applied for industrial chemicals manufacture from $CO_2$ in the future.

## Methods

For detailed information, please see Supplementary information.

**Plasmid construction.** Plasmids used in the study are listed in Supplementary Table 4. All plasmids were constructed by Gibson DNA assembly. The primers and DNA sequences are shown in the Supplementary Data 3.

**Protein synthesis and purification.** Acetate kinase and hexokinase were purchased from Sigma-Aldrich. The genes of ThDP-dependent enzymes and PKs were ligated into the expression vector pET-28a via *Nde*I and *Xho*I restriction sites, respectively. The remaining genes, AckA, PTA, were constructed in the same position in pET-28a. All genes were transformed into *E. coli* BL21 (DE3) for expression. Proteins were purified by His-Spin protein mini-prep columns (Zymo Research). The protein concentration was determined using a BCA Protein Assay Reagent Kit (Pierce, USA) with BSA as the standard.

**Protein engineering of glycolaldehyde synthase.** First, a single-point saturation mutation method was used to construct the clone library. The activity of the mutants were determined by measuring the amount of glycolaldehyde produced by the whole cell catalytic systems. Amino acids with highly active mutants were selected for subsequent iterative saturation mutations.

**Demonstration of the SACA pathway in vitro.** Formaldehyde as the initial substrate: The assay was set up at 37 °C in a final volume of 1 mL containing 50 mM potassium phosphate buffer (pH 7.5), 5 mM MgSO$_4$, 1 mM ThDP, and 2 mg mL$^{-1}$ GALS.

Glycolaldehyde as the initial substrate: The assay was set up at 37 °C in a final volume of 2 mL containing 50 mM potassium phosphate buffer (pH 7.5), 5 mM MgSO$_4$, 1 mM ThDP, 1 mM ADP, 0.2 mg mL$^{-1}$ acetate kinase, 10 U hexokinase, 20 mM glucose, 2 mg mL$^{-1}$ ACPS.

One-carbon assimilation pathway in vitro from formaldehyde to acetyl-CoA: The assay was set up at 37 °C in a final volume of 1 mL containing 50 mM triethanolamine buffer (pH 7.5), 5 mM MgSO$_4$, 1 mM ThDP, 1 g L$^{-1}$ formaldehyde, 20 mM CoA, 10 mM K$_3$PO$_4$, 2 mg mL$^{-1}$ GALS, 2 mg mL$^{-1}$ ACPS, 0.5 mg mL$^{-1}$ PTA.

**Analysis for ¹³C-labeled metabolites in clarified cellular lysates and in vivo.** In clarified cellular lysates: Recombinant plasmid, pET-28a-*GALS-ACPS-PTA*, was constructed using different enzyme cutting sites and transformed into *E. coli* BL21 (DE3) for expression. The reaction was conducted at 37 °C by adding 1 mM Coenzyme A (CoA) and 0.3 g L$^{-1}$ ¹³C-labeled formaldehyde in clarified cellular lysates. Sample was enriched by the cryoconcentration to 100 μL and detected by LC-MS. The procedures of various control experiments were consistent with the above.

In vivo: The strain containing bi-plasmid (*BsMDH*-pCDF, *GALS-ACPS-PTA*-pET-28a) or the strain containing bi-plasmid (*BsMDH*-pCDF, pET-28a) was used for ¹³C-labeled metabolic tracer analysis in vivo. The cells that had induced the protein in LB were re-inoculated into M9 medium containing 8 g L$^{-1}$ ¹³C-labeled methanol. Cells from different culture times were collected and used to detect intracellular metabolites and proteinogenic amino acids. Samples were concentrated and tested with LC-MS.

**Growth of *E. coli* strains.** Growth experiments were completed using LB medium with all combinations of −/+ formaldehyde (80 mg L$^{-1}$, − means the absence of substrate, + means the presence of substrate) and −/+ glycolaldehyde (0.4 g L$^{-1}$). Engineered *E. coli* strain, containing the recombinant plasmid,

GALS-ACPS-PTA-pET-28a, was incubated in 200 mL LB with appropriate resistance at 37 °C, 220 rpm. Cells were induced using IPTG with 0.5 mM as final concentration at mid-log phase (OD$_{600}$ ~0.6). After 1 h, formaldehyde or glycolaldehyde was added in cultures. And then, OD$_{600}$ was measured at each specified time point. Biomass of cellular dry weight (CDW) was determined using a conversion factor of 0.323 gCDW L$^{-1}$ OD$_{600}$$^{-1}$.

The strain containing bi-plasmid (BsMDH-pCDF, GALS-ACPS-PTA-pET-28a) or the strain containing bi-plasmid (BsMDH-pCDF, pET-28a) was used in the growth experiment with methanol as substrate. The strains were incubated in 200 mL LB with appropriate resistance at 37 °C, 220 rpm. Cells were induced using 0.5 mM IPTG at mid-log phase (OD$_{600}$ ~0.6). After 1 h, cells were transferred into M9 minimal medium supplemented with 1 g L$^{-1}$ Yeast extract, 2 g L$^{-1}$ Tryptone, 0.1 mM IPTG, trace elements (0.5 mg L$^{-1}$ FeCl$_3$•6H$_2$O, 0.09 mg L$^{-1}$ ZnSO$_4$•7H$_2$O, 0.088 mg L$^{-1}$ CuSO$_4$•5H$_2$O, 0.045 mg L$^{-1}$ MnCl$_2$, 0.09 mg L$^{-1}$ CoCl$_2$•6H$_2$O) and adequate resistance with or without 8 g L$^{-1}$ methanol. And then, OD$_{600}$ was measured at each specified time point.

**Reporting Summary**. Further information on experimental design is available in the Nature Research Reporting Summary linked to this article.

## Data availability

Authors confirm that all relevant data are included in the paper or its supplementary information files. The structure of GALS has been deposited at the PDB under accession code 6A50. Source Data are provided for Figs. 2–7 and Supplementary Figs. 3, 5–7, 12, 15, 17–24 as a Source Data file.

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

## Acknowledgements

We thank Prof. Yanhe Ma, Qinhong Wang, Jun Sun and Dr. Pengli Cai from Tianjin Institute of Industrial Biotechnology for reading the manuscript and for their helpful comments. We also thanks three anonymous reviewers who help us to improve this manuscript a lot. This work was supported by the Key Research Program of the Chinese Academy of Sciences (ZDRW-ZS-2016) to Y.L. and H.J. and by the 973 Program (2015CB755704) and Tianjin Science and Technology Plan Program (15PTCYSY00020) to H.J. and a National Natural Science Foundation of China (21773138) to Y. L.

## Author contributions

X.L., Yu.L., Y.Y., H.J. and Y.M. designed the SACA pathways; X.L., Yu.L., Y.Y., H.J., Yin. L., W.W., C.T., S.L. and Y.M. wrote the manuscript; Yu.L., Q.W., Z.Y., J.C., C.L. and J.L. contributed to the computer screening and structural analysis of enzymes; X.L., S.Y., J.G. and L.Y. contributed to the directed evolution of GALS; S.W., Y.G. and Y.N. contributed to the crystal of GALS; Yu.L., Yong.L., X.W. contributed to the analysis of the catalytic mechanism of ACPS; X.L., Y.Y., X.Y., H.L., T.C. and H.M. performed the experiments of synthesis of acetyl-CoA from formaldehyde in vitro; X.L. performed the experiments of 13C-labeling and growth experiments in vivo. L.L., Z.Z., B.Z. and Y.M. provided support for the detection of 13C-labeled metabolites.

## Additional information

**Competing interests:** The authors declare no competing interests.

