## [Peer Review File · Nature Communications]

Reviewers' Comments:

Reviewer #1:

Remarks to the Author:

Summary

This manuscript describes construction of a synthetic pathway for acetyl-CoA synthesis from formaldehyde. Through identification and extensive engineering of two separate enzymes found in nature, the authors demonstrated in vitro and in vivo conversion of formaldehyde to acetyl-CoA. *E. coli* was used as the host to demonstrate growth on formaldehyde using the synthetic pathway. Overall, this manuscript feels very short and could be expanded with more discussion and references. Though this concept is novel, I believe the current manuscript is missing essential experiments and appropriate controls that are required to clearly demonstrate the concept. Therefore, I recommend a major revision to address these concerns. The revision will require additional experiments and analysis

Major Comments

The title should be expanded to indicate that acetyl-CoA is synthesized from formaldehyde, a 'hub' metabolite of C1 metabolism.

Line 30: This statement seems misleading. These enzymes were not truly 'designed' but rather identified and repurposed through protein engineering. Perhaps the authors should consider rephrasing this statement.

Lines 48-52. "The most efficient pathway for the biosynthesis of acetyl-CoA in nature is the Wood-Ljungdahl (WL) pathway (6, 7)". Neither of these two refs make this statement or compares the WLP to other acetyl-CoA (or CO₂ fixation pathways). This was done by others much later. E.g., DOI: 10.1016/j.coche.2012.07.005 for CO₂ fixation (and indirectly for acetyl-CoA biosynthesis) but there may be other comparisons for acetyl-CoA.

Lines 52-56: These statements are incorrect and misleading. Not clear where in Ref. 9 is it stated what the authors write. The limitation of the WLP in producing large quantities of long-chain chemicals is not addressed in this ref. Other publications have dealt with that. E.g., DOI: 10.1016/j.coche.2012.07.005. If you want to criticize something, you should be specific and precise and say why so. But does the proposed pathway solve this problem? I am not sure it does, and in any case the amount of chemical energy in CH₂O is much higher than that in CO₂, so, the criticism and comparison is not fair. Significantly, the WLP can sustain biological life on CO₂ alone, and the trick is the energy generation through membrane-enabled processes. For the proposed pathway, it is not clear that it can sustain growth on the chemically much richer CH₂O (aerobic methylotrophs can, and those organisms and the associated pathways should be the basis for comparison of the proposed pathway). With regard to the 2nd statement and citing ref. 10, this is incorrect and misleading. Ref 10 and the earlier work from that lab have shown that the key WLP enzymes can be functionally expressed and that there are no issues with oxygen sensitivity. Although the complete WLP has not been functionally expressed in a heterologous host, the core and key individual WLP enzymes have been, and the rest is an issue of effort level and funding apparently.

This manuscript would benefit from an additional figure. Fig. 4A should be introduced as the first figure in the manuscript to illustrate the pathway. Expanding this figure would also be helpful to illustrate remaining central carbon metabolism from acetyl-CoA and how C1 substrates (e.g. methanol and CO₂) feed into formaldehyde. Cofactor utilization should also be included in the figure.

Fig. S3: The protein gels in this figure should be cleaned up if conclusions are to be made from

them. It is important to show clear expression in the soluble lysate since higher expression in the purified fractions could result from different purification efficiencies.

Line 98: This is a large library. It would be beneficial here to discuss the screening strategy in the main text to illustrate the high-throughput manner in which these variants were screened.

Table S1: I assume these parameters are for the substrate formaldehyde. If so, the K_m is extremely high (e.g. 59 mM at the lowest) compared to physiological formaldehyde concentrations (~tens of micromolar). The authors should include a discussion around how this disagreement may affect in vivo operation. Is it possible instead to select for variants having a lower K_m ? This should be discussed as well.

Fig. S9: This seems to be introduced without much explanation. Please add discussion around the crystal structure and number of chains for GALS in the main text if this is essential.

Fig. 1D: This is a poor cartoon illustration to demonstrate the binding pocket volume. Perhaps this could be replaced with Fig. S10, which is much more detailed to demonstrate this result.

Line 122 & Fig. 2B: The description in the main text disagrees with the figure regarding PK #s. Based on the figure, PK2 exhibits the highest activity. Please change the main text to reflect the numbers in the figure.

Line 129: It would be interesting to compare how this energy barrier value compares with other related enzymes or enzymes throughout central carbon metabolism?

Line 176: Does PTA need to be overexpressed to achieve the phenotype? Is the chromosomal copy insufficient? Was the experiment performed with overexpression of only GALS and ACPS?

Fig. S18: This figure is difficult to interpret and should be cleaned up. Perhaps the authors could remake this figure as a bar graph showing the relative abundances of the mass isotopomers (e.g. M+0, M+1, and M+2). Furthermore, this is an important figure to demonstrate the concept with ^{13}C tracing and should be included in the manuscript, not the supplemental material. It should also be specified explicitly whether this is in vitro or in vivo data. I assume in vitro.

Fig. 4B: If GALD is truly assimilated to support increased biomass, the authors should report the biomass yield on GALD (i.e. gDW/g). Two additional controls are required here as well. Both strains should be repeated in the absence of GALD to verify the biomass improvement is solely from GALD and not from strain differences. The figure legend should be more detailed to specify the rich medium and other experimental details, as to not mislead the reader.

Fig. 4C: The concentration of formaldehyde here is very high (~3 mM), which is extremely toxic to cells. It is not unlikely (in fact I think it is most likely) that the low growth extent of the red line (28a+ FALD) is due to FALD inhibition. One then needs an important control: the 28a strain in the rich medium without FALD. This will assess the impact FALD inhibition in the absence of FALD uptake. In this strain, they should assess the rate of FALD detoxification by 28a. The blue-line strain should also be examined in the absence of FALD. FALD consumption should also be determined here and is easily quantifiable via the colorimetric NASH reaction. This would allow determination of the biomass yield on formaldehyde (i.e. gDW/g). Finally, ^{13}C -formaldehyde should be used here to verify the contribution of formaldehyde-derived carbon in metabolites and amino acids. The improved growth could simply be from detoxification of formaldehyde via partial metabolism through the pathway. Did the authors examine lower formaldehyde concentrations here (e.g. more physiological concentrations on the order of micromolar)? If required, the authors should repeat this experiment using a fed-batch formaldehyde feeding strategy to maintain physiological formaldehyde concentrations below the toxicity level.

A critical experiment that is missing is examination for growth on formaldehyde alone in minimal media. This should be performed as this would be a novel and impactful demonstration. As above, a fed-batch strategy may be required to maintain low formaldehyde concentrations. If growth on formaldehyde alone is not possible, what is the authors hypothesis for this? Are there other limitations in the cell besides the synthetic pathway?

I'm not entirely sure what important features the X-ray structure document shows. Perhaps the authors could add more discussion around this in the main text if it is essential.

Minor Comments

Line 30: This is the first time SACA is introduced and should state explicitly what SACA stands for ('Synthetic Acetyl-CoA' pathway).

Fig. 1B: What does ND mean, not detectable or not determined? Please specify in the main text that this is in vitro data.

Line 99: Perhaps change 'residues' to 'mutations.'

Lines 109 & 125: S should be lowercase when describing the catalytic efficiency.

Fig. S16: Please indicate what the ~6.5 minute peak is. I assume this is formaldehyde.

Line 153: It is unclear why is this obvious. What side product would be made if ACPS uses formaldehyde as a substrate?

Line 176: This is the first time E. coli is introduced. The authors should introduce E. coli as the host organism earlier in the manuscript.

Line 199: Please rephrase this sentence for clarity. 'Level of chemical drive' is an odd way to state driving force.

Reviewer #2:

Remarks to the Author:

Remarks to authors:

In this manuscript (NCOMMS-18-18376-T), Liu et al described the construction of a synthetic pathway to produce one molecule of acetyl-CoA from two molecules of formaldehyde. First, a BFD enzyme (3FZN) was screened from ThDP-dependent enzymes from PDB. This enzyme was shown to convert FALD to GALD, and has a good expression level in E. coli. This enzyme was further engineered by saturation mutagenesis of the substrate binding pocket to improve its catalytic efficiency. Next, a phosphoketolase was selected and found to convert GALD to AcP. The authors performed simulation to propose a catalytic mechanism, but no further protein engineering was performed. By adding a known third enzyme PTA, the entire pathway was tested in vitro and the final yield of acetyl-CoA was claimed to be 33%. Finally, the pathway was tested in E.coli cell lysates and using ¹³C labeling, the authors detected TCA cycle intermediates derived from formaldehyde. The pathway was also shown to support E.coli growth by feeding GALD or formaldehyde. Overall, the study is novel and the in vitro evidence is strong. However, the in vivo evidence seems to be weak and it is hard to conclude that the current SACA pathway can support efficient production of acetyl CoA derived chemicals. It will greatly improve the significance and impact of this work if the following points are remedied.

Major points:

1. the data about improvement of catalytic efficiency reported in table s1 is a crucial evidence of successful engineering. Thus, it should be moved to Fig. 1, instead of in supplementary material. Preferably, curves showing change of kinetic parameters from M1 to M4 should be drawn for easier understanding, instead of showing a table.

2. Fig. 3D: fig. 3D is not adequately discussed in the text. It is not clear to me whether the two curves are from one sample (feeding CoA but also detected acetic acid) or from two experiments (feeding CoA vs No CoA added). In addition, the method to generate curves in Fig. 3B and 3D should be mentioned in the legend. Is there a specific fitting method?

3. Fig. 4A: the authors claim that the SACA pathway is orthogonal, which means that 13C can only flow from acetyl-CoA to the central metabolism. However, no data is shown to support this orthogonality. Necessary control experiments, for example, pathways lacking one of the three enzymes should be tested to see if 13C can flow into central metabolism from one of the intermediate products (GALD, AcP).

4. the 13C data in fig. S18 shows the carbon flow from FALD to TCA cycle. This is a crucial evidence showing the SACA pathway works in cell lysates. As a result, the data should be moved to fig. 4, instead of in supplementary materials.

5. in fig. S18ABCD, the blue lines show that the majority of measured TCA intermediates are not 13C2 labeled, which means they are not derived from FALD. It seems that the efficiency of producing acetyl-CoA from SACA pathway is very low, resulting in doubts whether SACA pathway can be practically useful to efficiently produce chemicals from FALD. It should be discussed in the text why this happens.

6. Fig. 4B and 4C: an additional control experiment showing the growth curves of both strains in growth media, without adding either GALD or FALD, should be performed. Ideally, the two strains should reach to the same OD after saturation, so that we know the growth advantage in fig 4B is not due to simply expression of the enzymes, but indeed due to the reactions they carry out.

7. Fig. 4B and 4C: Control strains lacking one of the three enzymes should also be constructed and tested. Ideally, they should perform the same as 28a. These control experiments will make the data more convincing.

8. line 122: from fig 2B, it should be PK1, 4, 8 that did not show significant difference. Also, in line 123, it should be PK2 that has the highest activity. Please clarify.

9. line 168: there is no data about yield in Fig. 3D. Not sure where the number "33%" come from.

10. according to line 287 in methods, the biosynthesis of acetyl CoA was not performed exactly "in vivo". Instead, it was performed in cell lysate. To show that SACA works in vivo, FALD has to be fed to live e.coli cells. After fermentation, cellular metabolites can then be analyzed. According to the current data, it is not convincing that SACA can work in live cells to produce acetyl-CoA from FALD, in an efficient manner that will be of interest to chemical/metabolic engineers.

11. it is not clear if the e.coli with the SACA pathway can grow with FALD as the sole carbon source. If not, the utility of the SACA pathway will be limited. The authors should discuss more about the current limitations of SACA and how we can overcome them to realize production of value added compounds using SACA, since no such example was demonstrated in the current manuscript. In addition, given its synthetic nature, SACA should also work in other microbial cell factories such as yeast. The authors should discuss the potential extensibility of SACA.

Minor points:

1. Fig. 2B: It should be explained in the legend the meanings of the numbers in the NJ tree. The

Activity of the PKs (x-axis) should also be explained in more detail. For example, it should be labeled as "AcP produced" instead of the vague "Activity". In addition, explanation about the activity unit (U/mg) should be added in the legend.

2. Fig. 2C: provide the full names of TS1, TS2 in the legend.

3. line 71: change "insensitive" to "sensitivity".

4. line 84: it should be explained in brief why candidates with short distances are more likely to perform the reaction? Not long distance?

5. line 88: the authors show that 3FZN and 4K9Q can produce GALD (fig. 1B). It came as a sudden result to me. The assay to test their activity should be mentioned before showing the result.

6. fig S4: in the legend the authors state that several mutations in BAL improve its activity. Data or a reference should be added to support this claim.

7. fig S5: the y-axis should be explained in detail what is the assay to measure the activity.

8. fig s8: the y-axis should be explained in detail what is the assay to measure the activity. Line 95: the authors state that 14 positions showed higher activities. However, it is hard to tell from fig s8 which 14 showed higher activities. It may be better to use box-whisker plot so that median activities can be compared with the double mutant?

9. fig. s9: it should be explained in the legend why GALS has two chains? Does it catalyze as a homodimer?

10. fig. s15: it was shown in the figure that H142 and G155 are important for ACPS. However, the roles of these two residues were not discussed in the text. Also, the y-axis should be explained in more detail in the legend.

11. line 151: the authors state that AcP would quickly degrade into acetic acid. A reference should be added to support their claim.

12. line 157: the relative concentrations of the two enzymes in the reaction mix should be mentioned.

13. line 160: the yield from GALD to AcP is missing. From fig 3B, only acetic acid concentration, not yield, was plotted.

14. fig s18C: why is one 13C fumaric acid also much higher than its natural intensity?

15. discussion about fig. S18E and F is missing from the text.

16. line 218: it is not clear how methanol can be used by the SACA pathway.

17. methods line 60: it is 4TKR in fig S2.

18. methods line 313: it is not clear whether IPTG was added when OD reaches 0.6 or the authors stopped adding IPTG when OD reaches 0.6. Please clarify.

Reviewer #3:

Remarks to the Author:

In this study, the authors describe the development of a new method to produce acetyl-CoA, using a three-step enzyme cascade that they term the synthetic acetyl-CoA or "SACA" pathway.

In the first part, the authors perform a high-throughput screen to mutagenize residues in the active site of a benzoylformate decarboxylase (BFD) to accommodate and condense two formaldehyde molecules to glyceolaldehyde. They solve the crystal structure of this designed glycerolaldehyde synthase (GALS) to visualize interactions with an intermediate analogue (IMA, which is in fact thiamine diphosphate (TPP)).

In the second part, the authors screen eight evolutionary distant Phosphoketolases (PKs) and identify an enzyme (which they term acetyl-phosphate synthase (ACPS)) that is capable of converting glycerolaldehyde to acetyl-phosphate (AcP).

In the final part, the authors construct the full SACA pathway by mixing GALS and ACPS in a single pot, together with known enzyme phosphate acetyltransferase (PTA, which converts AcP into acetyl-CoA) to achieve a ~33% reaction yield of acetyl-CoA production in vitro. They end by showing that the enzymes potentially produce acetyl-CoA in vivo, when co-expressed in E.coli.

Overall, except for the in vivo experiments, I believe the data supports the major claims.

However, my main issue with the manuscript lies in the reporting of the data. Besides the poor language, I find the data difficult and frustrating to interpret. This is largely due to a persistent problem is the lack of the explanation of assay rational, design, let alone assay conditions, standards and interpretations in both the main text and figure legends. Most information can be found in the supplementary data, but the reader is often left guessing based on brief descriptions in the manuscript. Additionally, most figure legends are far too minimalistic, and often do not properly describe or help interpreting the figure.

There are too many minor textual problems to list all here, so I'll refrain to the major issues:

- Line 83: How was the docking performed and based on what?
- Figure 1A: What is this structure? I also do not see the reaction mechanism from the figure legend.
- Figure S3, please remove non-transparent labels from gels.
- Line 91: what was the rational to screen these residues in BAL? Are enzymes related? Additionally, how does this related to the structure shown in S7 (the figure legend does not specify what this structure is).
- Figure S8: what do the different colors represent?
- Table S1: the first two mutants do not seem improvements. Are the mutations additive? Please include the data, so the data quality can be judged. How does this relate to the data in S5?
- Figure 1D is not useful (and difficult the read with purple on blue), perhaps the point can better be made with figure S10.
- Fig 2B, the numbering does not correspond with the numbering in the text.
- Line 120: please show gels
- Figure S12: are these averages?
- Line 146: why are these concentrations "high"? It is still well below the Km (otherwise the activity could not have increased linearly). The reported yield nicely agrees with the reported kinetic parameters for this assay time. In Figure 3C, the authors show that FALS is likely inhibiting the subsequent step in the SACA. Therefore, by extending the assay time, can the final FALS concentration be reduced further, potentially increasing the final yield of the full SACA pathway?
- There are no controls for the experiment in S18.
- The experiments in Fig4 give a preliminary suggestion that the SACA enzymes contribute to growth, but adding more data (like, different concentrations of substrate, adding controls like inactive or less active enzymes, measuring the decrease of FALS and GALS, etc.), would make this point stronger.

Reviewers' comments:

Reviewer #1 (Remarks to the Author):

Summary

This manuscript describes construction of a synthetic pathway for acetyl-CoA synthesis from formaldehyde. Through identification and extensive engineering of two separate enzymes found in nature, the authors demonstrated in vitro and in vivo conversion of formaldehyde to acetyl-CoA. *E. coli* was used as the host to demonstrate growth on formaldehyde using the synthetic pathway. Overall, this manuscript feels very short and could be expanded with more discussion and references. Though this concept is novel, I believe the current manuscript is missing essential experiments and appropriate controls that are required to clearly demonstrate the concept. Therefore, I recommend a major revision to address these concerns. The revision will require additional experiments and analysis

Response: Thanks for your positive and constructive comments. We have supplemented additional experiments and thoroughly revised our manuscript according to your suggestions.

Major Comments:

The title should be expanded to indicate that acetyl-CoA is synthesized from formaldehyde, a 'hub' metabolite of C1 metabolism.

Response: We agree and changed the title to “Constructing a Synthetic Pathway for Acetyl-Coenzyme A from One-Carbon through Enzyme Design”.

Line 30: This statement seems misleading. These enzymes were not truly 'designed' but rather identified and repurposed through protein engineering. Perhaps the authors should consider rephrasing this statement.

Response: We turned down our statement. For example, in the abstract, “...by employing two repurposed enzymes glycolaldehyde synthase and acetyl-phosphate synthase.”

Lines 48-52. “The most efficient pathway for the biosynthesis of acetyl-CoA in nature is the Wood-Ljungdahl (WL) pathway (6, 7)”. Neither of these two refs make this statement or compares the WLP to other acetyl-CoA (or CO₂ fixation pathways). This was done by others much later. E.g., DOI: 10.1016/j.coche.2012.07.005 for CO₂ fixation (and indirectly for acetyl-CoA biosynthesis) but there may be other comparisons for acetyl-CoA.

Response: We removed this sentence in the revised vision.

Lines 52-56: These statements are incorrect and misleading. Not clear where in Ref. 9 is it stated what the authors write. The limitation of the WLP in producing large quantities of long-chain chemicals is not addressed in this ref. Other publications have dealt with that. E.g., DOI: 10.1016/j.coche.2012.07.005. If you want to criticize something, you should be specific and precise and say why so. But does the proposed pathway solve this problem? I am not sure it does, and in any case the amount of chemical energy in CH₂O is much higher than that in CO₂, so, the criticism and comparison is not fair. Significantly, the WLP can sustain biological life on CO₂

alone, and the trick is the energy generation through membrane-enabled processes. For the proposed pathway, it is not clear that it can sustain growth on the chemically much richer CH₂O (aerobic methylotrophs can, and those organisms and the associated pathways should be the basis for comparison of the proposed pathway). With regard to the 2nd statement and citing ref. 10, this is incorrect and misleading. Ref 10 and the earlier work from that lab have shown that the key WLP enzymes can be functionally expressed and that there are no issues with oxygen sensitivity. Although the complete WLP has not been functionally expressed in a heterologous host, the core and key individual WLP enzymes have been, and the rest is an issue of effort level and funding apparently.

Response: We agree with you and deleted this part. In fact, our research provides a possible pathway for acetyl-CoA biosynthesis from formaldehyde. Therefore in the revision, we focused our statement on the design of novel pathway for acetyl-CoA biosynthesis.

This manuscript would benefit from an additional figure. Fig. 4A should be introduced as the first figure in the manuscript to illustrate the pathway. Expanding this figure would also be helpful to illustrate remaining central carbon metabolism from acetyl-CoA and how C1 substrates (e.g. methanol and CO₂) feed into formaldehyde. Cofactor utilization should also be included in the figure.

Response: Thanks a lot for your suggestion. We created a new figure (Fig. 1 in the revised vision).

Fig. S3: The protein gels in this figure should be cleaned up if conclusions are to be made from them. It is important to show clear expression in the soluble lysate since higher expression in the purified fractions could result from different purification efficiencies.

Response: We created a clear figure and moved it into Fig. 2c.

Line 98: This is a large library. It would be beneficial here to discuss the screening strategy in the main text to illustrate the high-throughput manner in which these variants were screened.

Response: We described the details of the screening strategy in Page 6.

Table S1: I assume these parameters are for the substrate formaldehyde. If so, the K_m is extremely high (e.g. 59 mM at the lowest) compared to physiological formaldehyde concentrations (~tens of micromolar). The authors should include a discussion around how this disagreement may affect *in vivo* operation. Is it possible instead to select for variants having a lower K_m? This should be discussed as well.

Response: Since the native substrate of the benzoylformate decarboxylases (BFD) is not formaldehyde, the catalytic activity is very low. Based on our enzyme engineering, we already improved the activity about 70 folds, which is about two times higher than the similarly designed enzyme (FLS). However, this enzyme is still not able to directly use formaldehyde as carbon source *in vivo*, due to the extremely high K_m as you indicated. We discussed this issue in Page 10.

Fig. S9: This seems to be introduced without much explanation. Please add discussion around the

crystal structure and number of chains for GALS in the main text if this is essential.

Response: We explained the crystal structure of GALS in Page 6.

Fig. 1D: This is a poor cartoon illustration to demonstrate the binding pocket volume. Perhaps this could be replaced with Fig. S10, which is much more detailed to demonstrate this result.

Response: The figure was changed as suggested (Fig. 3c).

Line 122 & Fig. 2B: The description in the main text disagrees with the figure regarding PK #s. Based on the figure, PK2 exhibits the highest activity. Please change the main text to reflect the numbers in the figure.

Response: We apologize for this mistake. The text was modified accordingly.

Line 129: It would be interesting to compare how this energy barrier value compares with other related enzymes or enzymes throughout central carbon metabolism?

Response: We compared the energy barrier with the strain energy during the formation of DHETHDP using F6P as substrate in the Page 7.

Line 176: Does PTA need to be overexpressed to achieve the phenotype? Is the chromosomal copy insufficient? Was the experiment performed with overexpression of only GALS and ACPS?

Response: Yes, PTA needs to be overexpressed. When only overexpressing GALS and ACPS, the relative abundance of the double ¹³C-labelled acetyl-CoA was not higher than that of the empty vector (Fig. S17).

Fig. S18: This figure is difficult to interpret and should be cleaned up. Perhaps the authors could remake this figure as a bar graph showing the relative abundances of the mass isotopomers (e.g. M+0, M+1, and M+2). Furthermore, this is an important figure to demonstrate the concept with ¹³C tracing and should be included in the manuscript, not the supplemental material. It should also be specified explicitly whether this is in vitro or in vivo data. I assume in vitro.

Response: We appreciate your constructive comments. This figure was reorganized as your suggestion (Fig. 6). As we mentioned in the figure legend, the experiments of ¹³C labelling were performed using cellular lysate.

Fig. 4B: If GALD is truly assimilated to support increased biomass, the authors should report the biomass yield on GALD (i.e. gDW/g). Two additional controls are required here as well Both strains should be repeated in the absence of GALD to verify the biomass improvement is solely from GALD and not from strain differences. The figure legend should be more detailed to specify the rich medium and other experimental details, as to not mislead the reader.

Response: We calculated the biomass yield from glycolaldehyde. The amount of biomass derived from glycolaldehyde was 0.681 ± 0.028 gCDW/g (Page 10). And the figure legend (now it is Fig. 7b) was revised.

Fig. 4C: The concentration of formaldehyde here is very high (~3 mM), which is extremely toxic to cells. It is not unlikely (in fact I think it is most likely) that the low growth extent of the red line (28a+ FALD) is due to FALD inhibition. One then needs an important control: the 28a strain in the

rich medium without FALD. This will assess the impact FALD inhibition in the absence of FALD uptake. In this strain, they should assess the rate of FALD detoxification by 28a. The blue-line strain should also be examined in the absence of FALD. FALD consumption should also be determined here and is easily quantifiable via the colorimetric NASH reaction. This would allow determination of the biomass yield on formaldehyde (i.e. gDW/g). Finally, ¹³C-formaldehyde should be used here to verify the contribution of formaldehyde-derived carbon in metabolites and amino acids. The improved growth could simply be from detoxification of formaldehyde via partial metabolism through the pathway. Did the authors examine lower formaldehyde concentrations here (e.g. more physiological concentrations on the order of micromolar)? If required, the authors should repeat this experiment using a fed-batch formaldehyde feeding strategy to maintain physiological formaldehyde concentrations below the toxicity level.

Response: We added the assays as you mentioned. Based on our current results, the SACA pathway is functional within cells when using glycolaldehyde as carbon source. In addition, the SACA pathway is also able to efficiently remove the toxicity of formaldehyde, but not efficient enough to support cell growth using formaldehyde as carbon source. The consumption of formaldehyde in 28a strain or in SACA pathway strain also has been detected via the colorimetric NASH reaction. The consumption rate of formaldehyde in the SACA pathway strain increased almost one fold than that in the 28a strain (Supplementary Fig. 19).

In order to reduce the toxicity of formaldehyde to cells, we added a methanol dehydrogenase (*BsMDH*) to maintain a continuously low concentration of formaldehyde *in vivo*. After 26 hours incubation, the value of OD₆₀₀ in the strain containing both *BsMDH* and SACA pathway is significantly higher than the strain without the SACA pathway (*P*-value=0.005, T-test). Our results indicated that the SACA pathway is feasible to contribute biomass.

A critical experiment that is missing is examination for growth on formaldehyde alone in minimal media. This should be performed as this would be a novel and impactful demonstration. As above, a fed-batch strategy may be required to maintain low formaldehyde concentrations. If growth on formaldehyde alone is not possible, what is the authors hypothesis for this? Are there other limitations in the cell besides the synthetic pathway?

Response: Currently, we observed that the SACA pathway ran well *in vitro*. The contribution of the SACA pathway *in vivo* is still very tiny due to the too low substrate affinities for both designed enzymes and the toxicity of formaldehyde. By introducing *BsMDH* that could slowly provide formaldehyde from methanol *in vivo*, we achieved a successful growth using the SACA pathway *in vivo*. In future, to facilitate the application of the SACA pathway within cells, it is accessible to decrease substrate affinities of both designed enzymes by protein engineering and to increase the tolerance of formaldehyde in host, like *Pichia pastoris*, whose peroxisomes allow to maintain formaldehyde at a higher concentration.

I'm not entirely sure what important features the X-ray structure document shows. Perhaps the

authors could add more discussion around this in the main text if it is essential.

Response: We described more details about the crystal structure of GALs in Page 6.

Minor Comments

Line 30: This is the first time SACA is introduced and should state explicitly what SACA stands for ('Synthetic Acetyl-CoA' pathway).

Response: The caption was added accordingly.

Fig. 1B: What does ND mean, not detectable or not determined? Please specify in the main text that this is in vitro data.

Response: The ND means that no glycolaldehyde was detected. The explanation was added accordingly.

Line 99: Perhaps change 'residues' to 'mutations.'

Response: The description was changed accordingly.

Lines 109 & 125: S should be lowercase when describing the catalytic efficiency.

Response: We changed these words accordingly.

Fig. S16: Please indicate what the ~6.5 minute peak is. I assume this is formaldehyde.

Response: It should be something from the solvent since we did not add formaldehyde in these reactions. The substrate is glycolaldehyde. We described the details in the legend of Fig. S15.

Line 153: It is unclear why is this obvious. What side product would be made if ACPS uses formaldehyde as a substrate?

Response: Nothing new was detected by HPLC when ACPS used formaldehyde as substrate and formaldehyde would partially inhibit ACPS activity.

Line 176: This is the first time E. coli is introduced. The authors should introduce E. coli as the host organism earlier in the manuscript.

Response: It was introduced earlier accordingly (Page 3).

Line 199: Please rephrase this sentence for clarity. 'Level of chemical drive' is an odd way to state driving force.

Response: The sentence was rephrased accordingly (Page 4).

Reviewer #2 (Remarks to the Author):

Remarks to authors:

In this manuscript (NCOMMS-18-18376-T), Liu et al described the construction of a synthetic pathway to produce one molecule of acetyl-CoA from two molecules of formaldehyde. First, a BFD enzyme (3FZN) was screened from ThDP-dependent enzymes from PDB. This enzyme was

shown to convert FALD to GALD, and has a good expression level in *E. coli*. This enzyme was further engineered by saturation mutagenesis of the substrate binding pocket to improve its catalytic efficiency. Next, a phosphoketolase was selected and found to convert GALD to AcP. The authors performed simulation to propose a catalytic mechanism, but no further protein engineering was performed. By adding a known third enzyme PTA, the entire pathway was tested *in vitro* and the final yield of acetyl-CoA was claimed to be 33%. Finally, the pathway was tested in *E. coli* cell lysates and using ¹³C labeling, the authors detected TCA cycle intermediates derived from formaldehyde. The pathway was also shown to support *E. coli* growth by feeding GALD or formaldehyde. Overall, the study is novel and the *in vitro* evidence is strong. However, the *in vivo* evidence seems to be weak and it is hard to conclude that the current SACA pathway can support efficient production of acetyl CoA derived chemicals. It will greatly improve the significance and impact of this work if the following points are remedied.

Response: Thanks for your positive comments and suggestions. In this revision, we added more *in vivo* experiments to support our results.

Major points:

1. the data about improvement of catalytic efficiency reported in table S1 is a crucial evidence of successful engineering. Thus, it should be moved to Fig. 1, instead of in supplementary material. Preferably, curves showing change of kinetic parameters from M1 to M4 should be drawn for easier understanding, instead of showing a table.

Response: We created a new figure as suggested in Fig. 3a.

2. Fig. 3D: Fig. 3D is not adequately discussed in the text. It is not clear to me whether the two curves are from one sample (feeding CoA but also detected acetic acid) or from two experiments (feeding CoA vs No CoA added). In addition, the method to generate curves in Fig. 3B and 3D should be mentioned in the legend. Is there a specific fitting method?

Response: The two curves are from different experiments (CoA feeding vs without CoA feeding). We added the details in the legend (now it is Fig. 5d) accordingly.

3. Fig. 4A: the authors claim that the SACA pathway is orthogonal, which means that ¹³C can only flow from acetyl-CoA to the central metabolism. However, no data is shown to support this orthogonality. Necessary control experiments, for example, pathways lacking one of the three enzymes should be tested to see if ¹³C can flow into central metabolism from one of the intermediate products (GALD, AcP).

Response: Considering the high activity of formaldehyde, we are not sure if formaldehyde would react with other compounds in cell. Thus we removed the statement of orthogonality. But in our experiments, we did not observe significantly more the double ¹³C-labeled acetyl-CoA in the strains lacking ACPS and PTA than that in the 28a strain (Fig. S17). Therefore most of acetyl-CoA we observed should be synthesized by the SACA pathway.

4. the ¹³C data in Fig. S18 shows the carbon flow from FALD to TCA cycle. This is a crucial evidence showing the SACA pathway works in cell lysates. As a result, the data should be moved

to fig. 4, instead of in supplementary materials.

Response: Fig. S18 was moved to the main text in Fig. 6 as suggested.

5. in fig. S18ABCD, the blue lines show that the majority of measured TCA intermediates are not ¹³C₂ labeled, which means they are not derived from FALD. It seems that the efficiency of producing acetyl-CoA from SACA pathway is very low, resulting in doubts whether SACA pathway can be practically useful to efficiently produce chemicals from FALD. It should be discussed in the text why this happens.

Response: Indeed, the efficiency of SACA pathway is very low right now since the designed enzymes aren't as efficient as natural enzymes. We discussed this issue in Page11.

7. Fig. 4B and 4C: Control strains lacking one of the three enzymes should also be constructed and tested. Ideally, they should perform the same as 28a. These control experiments will make the data more convincing.

Response: The control experiments were added as suggested (Fig. 7 and Fig. S18).

8. line 122: from fig 2B, it should be PK1, 4, 8 that did not show significant difference. Also, in line 123, it should be PK2 that has the highest activity. Please clarify.

Response: We apologize for the mistake. The number of PKs was corrected accordingly in the revised main text.

9. line 168: there is no data about yield in Fig. 3D. Not sure where the number "33%" come from.

Response: The substrate of formaldehyde is 33.3 mM and the product of acetyl-CoA is 5.5 mM. One mole of acetyl-CoA requires two mole of formaldehyde. So the yield of acetyl-CoA from formaldehyde was about 33%. We added the details in Page9 accordingly.

6. Fig. 4B and 4C: an additional control experiment showing the growth curves of both strains in growth media, without adding either GALD or FALD, should be performed. Ideally, the two strains should reach to the same OD after saturation, so that we know the growth advantage in fig 4B is not due to simply expression of the enzymes, but indeed due to the reactions they carry out.

10. according to line 287 in methods, the biosynthesis of acetyl CoA was not performed exactly "in vivo". Instead, it was performed in cell lysate. To show that SACA works in vivo, FALD has to be fed to live e. coli cells. After fermentation, cellular metabolites can then be analyzed. According to the current data, it is not convincing that SACA can work in live cells to produce acetyl-CoA from FALD, in an efficient manner that will be of interest to chemical/metabolic engineers.

11. it is not clear if the e. coli with the SACA pathway can grow with FALD as the sole carbon source. If not, the utility of the SACA pathway will be limited. The authors should discuss more about the current limitations of SACA and how we can overcome them to realize production of value added compounds using SACA, since no such example was demonstrated in the current manuscript. In addition, given its synthetic nature, SACA should also work in other microbial cell factories such as yeast. The authors should discuss the potential extensibility of SACA.

Response: Since question 6, 10 and 11 are related, we putted them together.

Based on our current results, the SACA pathway is functional within cells using glycolaldehyde as carbon source. The engineered strain containing the SACA pathway with more than 0.4 g/L glycolaldehyde supply has remarkable higher OD₆₀₀ than those strains without glycolaldehyde supply or the SACA pathway (Fig. 7b). The contribution of glycolaldehyde on biomass reached to 0.681 ± 0.028 gCDW/g glycolaldehyde.

Although the SACA pathway is not efficient enough for cell growth using formaldehyde as carbon source, the consumption rate of formaldehyde in the SACA pathway strain increased almost one fold than that in the 28a strain (Supplementary Fig. S19).

Furthermore, by introducing the methanol dehydrogenase (*BsMDH*) that would gradually produce formaldehyde *in vivo*, we found that the strain containing both *BsMDH* and the SACA pathway has significantly higher OD₆₀₀ than those controls without methanol supply or the SACA pathway (Fig. 7d). Thus our results clearly documented that the SACA pathway can produce biomass using glycolaldehyde and continuously low amount of formaldehyde as supplemental carbon sources.

Minor points:

1. Fig. 2B: It should be explained in the legend the meanings of the numbers in the NJ tree. The Activity of the PKs (x-axis) should also be explained in more detail. For example, it should be labeled as “AcP produced” instead of the vague “Activity”. In addition, explanation about the activity unit (U/mg) should be added in the legend.

Response: The x-axis label and explanation in the legend were changed (now it is Fig. 4b).

2. Fig. 2C: provide the full names of TS1, TS2 in the legend.

Response: The full names of TS1, TS2 were added in the legend accordingly (now it is Fig. 4c).

3. line 71: change “insensitive” to “sensitivity”.

Response: The sentence has been paraphrased.

4. line 84: it should be explained in brief why candidates with short distances are more likely to perform the reaction? Not long distance?

Response: In the catalytic mechanism of ThDP-dependent enzymes, C2 atom in ThDP is an active center and the distance between C2 atom in TPP and glycolaldehyde is critical for triggering the catalytic reaction. The distance is usually used as a criterion for judging whether the reaction could initiate. Short distance means that there is a greater chance for C2 atom to attack the ligand and the reaction can initiate easily, but it is the opposite when the distance is large.

5. line 88: the authors show that 3FZN and 4K9Q can produce GALD (fig. 1B). It came as a sudden

result to me. The assay to test their activity should be mentioned before showing the result.

Response: The details about the assays have been described in the main text (Page 5).

6. fig S4: in the legend the authors state that several mutations in BAL improve its activity. Data or a reference should be added to support this claim.

Response: A reference was added in Page 5.

7. fig S5: the y-axis should be explained in detail what is the assay to measure the activity.

Response: The details have been added to the legend (now it is Fig. S3) as suggested.

8. fig s8: the y-axis should be explained in detail what is the assay to measure the activity. Line 95: the authors state that 14 positions showed higher activities. However, it is hard to tell from fig s8 which 14 showed higher activities. It may be better to use box-whisker plot so that median activities can be compared with the double mutant?

Response: 14 positions were highlighted by the red pane in the revised Fig. S6. The y-axis was explained in the legend.

9. fig. s9: it should be explained in the legend why GALS has two chains? Does it catalyze as a homodimer?

Response: Yes, GALS plays function as a homodimer. We added the details about the structure of GALS in the revision.

10. fig. s15: it was shown in the figure that H142 and G155 are important for ACPS. However, the roles of these two residues were not discussed in the text. Also, the y-axis should be explained in more detail in the legend.

Response: The roles of these two residues were discussed in the text accordingly, and the y-axis was explained in detail in the legend (now it is Fig. S14) as suggested.

11. line 151: the authors state that AcP would quickly degrade into acetic acid. A reference should be added to support their claim.

Response: The reference was added accordingly.

12. line 157: the relative concentrations of the two enzymes in the reaction mix should be mentioned.

Response: The concentrations of the two enzymes in the reaction mix were added in the legend of Fig. 5.

13. line 160: the yield from GALD to AcP is missing. From fig 3B, only acetic acid concentration, not yield, was plotted.

Response: The y-axis (now it is Fig. 5b) has been changed to the yield.

14. fig s18C: why is one ¹³C fumaric acid also much higher than its natural intensity?

Response: The peak should not be fumaric acid since the molecular weight (116.0736) of this compound is larger than fumaric acid (116.007) about 65 ppm, which is higher than the system error (less than 10 ppm).

15. discussion about fig. S18E and F is missing from the text.

Response: We created a new Fig. 6.

16. line 218: it is not clear how methanol can be used by the SACA pathway.

Response: Methanol can be oxidized to formaldehyde and then used by the SACA pathway (Fig. 1a). In this vision, we introduced the methanol dehydrogenase (*BsMDH*) into the strain containing the SACA pathway. The total biomass in this strain is significantly higher than that in the strain without the SACA pathway. It is feasible to use methanol by the SACA pathway.

17. methods line 60: it is 4TKR in fig S2.

Response: We apologize for this mistake. It was changed.

18. methods line 313: it is not clear whether IPTG was added when OD reaches 0.6 or the authors stopped adding IPTG when OD reaches 0.6. Please clarify.

Response: We revised the methods part. IPTG was added when OD reached 0.6.

Reviewer #3 (Remarks to the Author):

In this study, the authors describe the development of a new method to produce acetyl-CoA, using a three-step enzyme cascade that they term the synthetic acetyl-CoA or “SACA” pathway. In the first part, the authors perform a high-throughput screen to mutagenize residues in the active site of a benzoylformate decarboxylase (BFD) to accommodate and condense two formaldehyde molecules to glyceolaldehyde. They solve the crystal structure of this designed glycerolaldehyde synthase (GALS) to visualize interactions with an intermediate analogue (IMA, which is in fact thiamine diphosphate (TPP)).

In the second part, the authors screen eight evolutionary distant Phosphoketolases (PKs) and identify an enzyme (which they term acetyl-phosphate synthase (ACPS)) that is capable of converting glycerolaldehyde to acetyl-phosphate (AcP).

In the final part, the authors construct the full SACA pathway by mixing GALS and ACPS in a single pot, together with known enzyme phosphate acetyltransferase (PTA, which converts AcP into acetyl-CoA) to achieve a ~33% reaction yield of acetyl-CoA production in vitro. They end by showing that the enzymes potentially produce acetyl-CoA in vivo, when co-expressed in *E.coli*. Overall, except for the in vivo experiments, I believe the data supports the major claims.

However, my main issue with the manuscript lies in the reporting of the data. Besides the poor language, I find the data difficult and frustrating to interpret. This is largely due to a persistent problem is the lack of the explanation of assay rational, design, let alone assay conditions, standards and interpretations in both the main text and figure legends. Most information can be found in the supplementary data, but the reader is often left guessing based on brief descriptions

in the manuscript. Additionally, most figure legends are far too minimalistic, and often do not properly describe or help interpreting the figure.

Response: Thank you very much for your positive and constructive comments. In this vision, we extended the main text with more assays *in vivo* and adjusted many important results in the supplementary data to the main text. At the same, we carefully revised our writing in the main text and figure legends.

There are too many minor textual problems to list all here, so I'll refrain to the major issues:

- Line 83: How was the docking performed and based on what?

Response: The docking experiment was performed with rosettaligand module in ROSETTA suite. Through the docking experiment, we placed the substrate into the catalytic pocket according to the rosetta substrate binding energy (interface delta_X). We attached the docking protocol in this vision.

- Figure 1A: What is this structure? I also do not see the reaction mechanism from the figure legend.

Response: It is the theozyme model, which include glycolaldehyde and it's interaction with the cofactor ThDP. We described more about the reaction mechanism in Page 5.

- Figure S3, please remove non-transparent labels from gels.

Response: We cleaned up the protein gels and moved this figure into Fig. 2c.

- Line 91: what was the rational to screen these residues in BAL? Are enzymes related? Additionally, how does this related to the structure shown in S7 (the figure legend does not specify what this structure is).

Response: It is reported that these residues in BAL can increase the activity of formaldehyde condensation. In BFD, there are some corresponding sites with BAL (Fig. S2). Experimental results showed that W86R-N87T mutant in BFD also significantly improved catalytic activity (Fig. S3).

Firstly we introduced W86R-N87T mutant into BFD. And then in order to further improve enzyme activity, we chose all residues within 8Å distance from the active center to do single-point saturation mutation (Fig. S4, which was S7 previously). We described the details of protein engineering in Page 6.

- Figure S8: what do the different colors represent?

Response: The single-point saturation mutation experiments were carried out in the 96-well plate, and the points of different colors represented the data from the different positions in the 96-well plate. We added the details in the figure legend (now it is Fig. S6).

- Table S1: the first two mutants do not seem improvements. Are the mutations additive? Please include the data, so the data quality can be judged. How does this relate to the data in S5?

Response: The kinetic parameter of mutant W86R-N87T in Table S1 (now it is

Fig 3a) was the initial reaction rate. It is different from we did in the Fig S5 (now it is Fig. S3), which is the yield of glycolaldehyde from whole-cell catalytic reaction. 10 OD cells in 1mL lysis buffer was used to test their activity by adding 2 g/L formaldehyde. We added the details of kinetic parameter in Fig. S7.

• Figure 1D is not useful (and difficult the read with purple on blue), perhaps the point can better be made with figure S10.

Response: The Figure 1D was changed as suggested (as Fig. 3c).

• Fig 2B, the numbering does not correspond with the numbering in the text.

Response: It was changed in the revision as suggested.

• Line 120: please show gels.

Response: The protein gels were added as Fig. S10.

• Figure S12: are these averages?

Response: Yes, these are averages. The error bars were added in the revised Fig. S11.

• Line 146: why are these concentrations “high” ? It is still well below the K_m (otherwise the activity could not have increased linearly). The reported yield nicely agrees with the reported kinetic parameters for this assay time. In Figure 3C, the authors show that FALS is likely inhibiting the subsequent step in the SACA. Therefore, by extending the assay time, can the final FALS concentration be reduced further, potentially increasing the final yield of the full SACA pathway?

Response: We did the experiments overnight (now it is Fig. 5c). The final yield of acetic acid in the system with the SACA pathway is even higher than that in the reaction system from glycolaldehyde to AcP with the same amount of formaldehyde (Fig. 5b, grey-line). This would be caused by partially releasing the function of ACPS since formaldehyde was gradually consumed by GALS.

• There are no controls for the experiment in S18.

Response: The control experiments have been added and the data has been updated in Fig. 6.

• The experiments in Fig4 give a preliminary suggestion that the SACA enzymes contribute to growth, but adding more data (like, different concentrations of substrate, adding controls like inactive or less active enzymes, measuring the decrease of FALS and GALS, etc.), would make this point stronger.

Response: We did more assays to make this point stronger. These results were described in the part “Confirmation of the SACA pathway *in vivo*”.

Firstly, by adding different concentrations of glycolaldehyde (Supplementary Fig. S18), the engineered strain that contains the SACA pathway with more than 0.4 g/L glycolaldehyde supply has remarkable higher OD_{600} than those strains without glycolaldehyde or the SACA pathway (Fig. 7b). The contribution of glycolaldehyde to biomass was 0.681 ± 0.028 gCDW/g

glycolaldehyde.

Secondly, when we tested cell growth using formaldehyde as supplemental carbon source, the strain with the SACA pathway was inhibited initially and then grew up normally. Due to the toxicity of formaldehyde, we could not input too many formaldehyde. The strain with the SACA pathway did not have more biomass with formaldehyde supply than those without formaldehyde.

Finally in order to reduce the toxicity of formaldehyde to cells, we added a methanol dehydrogenase (*BsMDH*) to maintain a continuously low concentration of formaldehyde *in vivo*. After 26 hours incubation, the value of OD₆₀₀ in the strain containing both *BsMDH* and SACA pathway is significantly higher than that in the strain without the SACA pathway (*P*-value=0.005, T-test). Our results indicated that the SACA pathway is able to contribute on biomass if formaldehyde is supplied continuously at low concentration.

Reviewers' Comments:

Reviewer #1:

Remarks to the Author:

This ms details construction of a novel pathway for the condensation of two formaldehyde molecules (which can be derived from methane, methanol, formic acid, and CO₂) into one acetyl-CoA molecule and is a revised version of a previously submitted ms.

Although the revised ms is much improved compared to the original submission, I still have one major and several minor concerns that should be addressed prior to acceptance. These concerns are detailed below.

Although the authors included more experiments in the revised ms, the in vivo results are still modest at best, which is common and expected for this type of work, so no concern here. However, this therefore requires a more comprehensive validation of the engineered pathway. For example, the authors did a great job to include a new experiment that suggests methanol utilization via the novel SACA pathway to improve growth of *E. coli* in vivo. However, the growth improvement seems to be less than 10%, so further experiments are warranted to validate the cells are actually incorporating methanol-derived carbon into central metabolites and biomass components to support the improvement in cell growth. This is a simple experiment to perform provided the analytics are available, which they seem to be based on ¹³C tracing experiments already included in the ms. The authors should repeat the in vivo growth assay with ¹³C labeled methanol and analyze samples at the end of growth for ¹³C labeling in central metabolites and biomass components. The control strain (BsMDH-28a) should also be analyzed in the presence of ¹³C labeled methanol to serve as a negative control.

Apart from the major concern above, I have several minor concerns that should be addressed.

Fig. 1b – I suspect the FLS pathway could have a yield of 100% if combined with NOG.

Fig. 6 and Fig. S17 – This is an incorrect way to represent the mass isotopomer distribution. The total of all mass isotopomers should equal 100%, i.e. $\sum(m+n) = 100\%$, $n=1,2,\text{etc.}$. Thus, $m+0$ should not be 100% for all figures. This needs to be corrected.

The authors should also consider correcting for natural ¹³C abundance in Fig. 6, Fig. S17, and additional ¹³C labeling data that may be included. This way, the controls will be around 0% and the improvement in $m+2$ will be more obvious.

Fig. 6a – Can the authors comment on why $m+1$ is decreased in SACA(+¹³C-FALD)? This result does not seem intuitive.

Figs. 6b-c – Did the authors detect any higher order mass isotopomers, i.e. $m+3$ or $m+4$, due to cycling of the TCA cycle? Can the authors comment on why not if so?

For all growth experiments, consumption profiles should be shown for all supplemental carbon sources, e.g. GALD, FALD, methanol, if biomass yields on these substrates are to be reported.

Line 307 (p. 10) – The biomass yield on methanol of 0.03 g/g should be compared to previously reported values to provide an idea of how the SACA pathway compares with other methanol utilization pathways that have been implemented in *E. coli* and other non-native methylotrophs.

Reviewer #2:

Remarks to the Author:

In this revision (NCOMMS-18-18376A), more control experiments (fig 7b, 7c, 7d) were performed to demonstrate that FALD can support growth of *E. coli* cells through the SACA pathway. Yet, no ¹³C experiments were performed to demonstrate in vivo carbon flow from FALD to acetyl-CoA, only in cell lysates. The authors also discussed that the current SACA efficiency is low for in vivo applications, due to the low substrate affinity of the two engineered enzymes. Overall, the SACA pathway is a novel pathway for one-carbon assimilation, yet its in vivo performance is not satisfactory. Apparently, further enzyme engineering is warranted for feasible in vivo applications such as fermentation.

Minor points:

1. In the revised fig 5b and 5d, it is still not mentioned how the connection lines were generated. In addition, the connection lines in fig 7d are straight, which are not consistent with fig 7b and 7c.
2. For fig s6, it is still not clear to this review how the data was presented. It was mentioned in the legend that the color dots represent data from different positions of the 96-well plate. Are they 96 technical replicates? If yes, the errors seem big and it would be useful to also show the average. If not, are they mutants of different amino acid residues of each position?
3. Fig 1a: molecular structures should also be shown for better understanding.
4. Fig 2c: the arrows should be drawn horizontally for more precise annotation. Currently, it is hard to tell which protein band in lane 2 the arrow points to.
5. Line 33: change "has improved" to "was improved".
6. Line 145: change "located" to "was located".
7. Line 243: change "was been gradually consuming" to "was gradually consumed".
8. Line 337: change "decrease" to "increase"?
9. Line 487: change "pane" to "panel".
10. Line 532: delete "was".
11. Line 579: the OD600 was measured, not by HPLC.
12. Related to my previous major point 7: the said control experiments were not added in either fig 7 or supplementary figures.

Reviewer #3:

Remarks to the Author:

I thank the authors for their revisions and am satisfied that they have answered my criticisms.

Reviewers' comments:

Reviewer #1 (Remarks to the Author):

This ms details construction of a novel pathway for the condensation of two formaldehyde molecules (which can be derived from methane, methanol, formic acid, and CO₂) into one acetyl-CoA molecule and is a revised version of a previously submitted ms.

Although the revised ms is much improved compared to the original submission, I still have one major and several minor concerns that should be addressed prior to acceptance. These concerns are detailed below.

Response: Thank you very much for your positive comments. We revised our MS again point-by-point according to your suggestions.

Although the authors included more experiments in the revised ms, the *in vivo* results are still modest at best, which is common and expected for this type of work, so no concern here. However, this therefore requires a more comprehensive validation of the engineered pathway. For example, the authors did a great job to include a new experiment that suggests methanol utilization via the novel SACA pathway to improve growth of *E. coli* *in vivo*. However, the growth improvement seems to be less than 10%, so further experiments are warranted to validate the cells are actually incorporating methanol-derived carbon into central metabolites and biomass components to support the improvement in cell growth. This is a simple experiment to perform provided the analytics are available, which they seem to be based on ¹³C tracing experiments already included in the ms. The authors should repeat the *in vivo* growth assay with ¹³C labeled methanol and analyze samples at the end of growth for ¹³C labeling in central metabolites and biomass components. The control strain (BsMDH-28a) should also be analyzed in the presence of ¹³C labeled methanol to serve as a negative control.

Response: Thanks a lot for your suggestion. We repeated the growth assays with ¹³C-labeled methanol and analyzed metabolites (Fig. 7). Comparing with the strain BsMDH-28a, in the strain BsMDH-SACA, we found significantly more double ¹³C-labeled aspartate and glutamate, which would derive from the TCA cycle. We also detected significantly more ¹³C-labeled glucose and phosphoenolpyruvate, which would be generated by the gluconeogenesis pathway using double ¹³C-labeled oxaloacetate, in BsMDH-SACA than those in BsMDH-28a. Thus, our results showed that formaldehyde that was generated from methanol, could be transformed into biomass by the SACA pathway *in vivo* (Page 9).

Apart from the major concern above, I have several minor concerns that should be addressed.

Fig. 1b – I suspect the FLS pathway could have a yield of 100% if combined with NOG.

Response: Certainly, the FLS pathway should have a yield of 100% via combining with NOG. Here we only obtained carbon yield of the FLS pathway from the published papers (Siegel et al., 2015, PNAS).

Fig. 6 and Fig. S17 – This is an incorrect way to represent the mass isotopomer distribution. The total of all mass isotopomers should equal 100%, i.e. $\sum (m+n) = 100\%$, $n=1,2,\text{etc.}$ Thus,

m+0 should not be 100% for all figures. This needs to be corrected.

Response: Thanks for your positive and constructive comments. We corrected all ¹³C labeled results by setting the total isotopomers as 100% (Supplementary Fig. 17 & 18 & Fig. 6c).

The authors should also consider correcting for natural ¹³C abundance in Fig. 6, Fig. S17, and additional ¹³C labeling data that may be included. This way, the controls will be around 0% and the improvement in m+2 will be more obvious.

Response: Thank you very much. In this revision, we only showed the results of double ¹³C labeled compounds in the text (Fig. 6b). There are significant differences for the tested double ¹³C labeled compounds between the strains with and without the SACA pathway (Page 9).

Fig. 6a – Can the authors comment on why m+1 is decreased in SACA(+¹³C-FALD)? This result does not seem intuitive.

Response: We corrected ¹³C labeled results by setting the total mass isotopomers as 100% (Supplementary Fig. 17). The decreased fraction of m+1 would be caused by the increased fraction of m+2.

Figs. 6b-c – Did the authors detect any higher order mass isotopomers, i.e. m+3 or m+4, due to cycling of the TCA cycle? Can the authors comment on why not if so?

Response: We did not detect significant differences in higher order mass isotopomers, i.e. m+3, m+4. It would be caused by the low amount of m+2 isotopomers in the TCA cycle (Page 9, Line 260).

For all growth experiments, consumption profiles should be shown for all supplemental carbon sources, e.g. GALD, FALD, methanol, if biomass yields on these substrates are to be reported.

Response: We added the consuming profiles of the supplemental carbon sources as you suggested (Supplementary Fig. 20, 21, & 22).

Line 307 (p. 10) – The biomass yield on methanol of 0.03 g/g should be compared to previously reported values to provide an idea of how the SACA pathway compares with other methanol utilization pathways that have been implemented in E. coli and other non-native methylotrophs.

Response: The corresponding discussion has been added (Page 11, Line 331).

Reviewer #2 (Remarks to the Author):

In this revision (NCOMMS-18-18376A), more control experiments (fig 7b, 7c, 7d) were performed to demonstrate that FALD can support growth of E. coli cells through the SACA pathway. Yet, no ¹³C experiments were performed to demonstrate in vivo carbon flow from FALD to acetyl-CoA, only in cell lysates. The authors also discussed that the current SACA efficiency is low for in vivo applications, due to the low substrate affinity of the two engineered enzymes. Overall, the SACA pathway is a novel pathway for one-carbon assimilation, yet its in vivo performance is not satisfactory. Apparently, further enzyme engineering is warranted for feasible in vivo applications

such as fermentation.

Response: Thank you for your positive and constructive comments. We added ^{13}C -labeled assays *in vivo*. We detected significantly more double ^{13}C -labeled compounds such as aspartate, glutamate, glucose and phosphoenolpyruvate. These results indicated that the SACA pathway is workable *in vivo* (Page 9).

Minor points:

1. In the revised fig 5b and 5d, it is still not mentioned how the connection lines were generated. In addition, the connection lines in fig 7d are straight, which are not consistent with fig 7b and 7c.

Response: In fig. 5b&5d, points on each line represent yields of acetic acid or acetyl-CoA at different time points. We added the details in the revised figure legend. In fig. 7, the figure has been changed as suggestion.

2. For fig s6, it is still not clear to this review how the data was presented. It was mentioned in the legend that the color dots represent data from different positions of the 96-well plate. Are they 96 technical replicates? If yes, the errors seem big and it would be useful to also show the average. If not, are they mutants of different amino acid residues of each position?

Response: Here, the x-axis label represents different positions in BFD. We did saturation mutation in each position. For each position, we randomly picked 90 mutants. The y-axis label represents the relative catalytic activity of different mutants. The figure was modified and the corresponding description was added in the legend.

3. Fig 1a: molecular structures should also be shown for better understanding.

Response: This figure was changed as your suggestion (Fig. 1a).

4. Fig 2c: the arrows should be drawn horizontally for more precise annotation. Currently, it is hard to tell which protein band in lane 2 the arrow points to.

Response: The figure was changed as suggested (Fig. 2c).

5. Line 33: change “has improved” to “was improved” .

Response: The sentence has been paraphrased (Page 2, Line 32).

6. Line 145: change “located” to “was located” .

Response: The sentence has been paraphrased (Page 5, Line 142).

7. Line 243: change “was been gradually consuming” to “was gradually consumed” .

Response: The sentence has been paraphrased (Page 8, Line 238).

8. Line 337: change “decrease” to “increase” ?

Response: The sentence has been paraphrased (Page 11, Line 337).

9. Line 487: change “pane” to “panel” .

Response: The sentence has been paraphrased (Page 17, Line 484).

10. Line 532: delete “was” .

Response: The sentence has been changed.

11. Line 579: the OD600 was measured, not by HPLC.

Response: We apologize for the mistake. The sentence was changed.

12. Related to my previous major point 7: the said control experiments were not added in either fig 7 or supplementary figures.

Response: We appreciate your constructive comments. We added the control experiments in Fig. 7 and Supplementary Fig. 18 as suggestion.

Reviewer #3 (Remarks to the Author):

I thank the authors for their revisions and am satisfied that they have answered my criticisms.

Response: Thank you very much for your positive and constructive comments.

Reviewers' Comments:

Reviewer #1:

Remarks to the Author:

The revision with the additional data addressed all remaining concerns.

Still, prior to publication, error bars should be added for the ^{13}C labeling experiments per standard expectations:

Fig. 6C and Suppl. fig 17.

REVIEWERS' COMMENTS:

Reviewer #1 (Remarks to the Author):

The revision with the additional data addressed all remaining concerns. Still, prior to publication, error bars should be added for the ¹³C labeling experiments per standard expectations: Fig. 6C and Suppl. fig 17.

Response: Thanks a lot for your suggestion. We added error bars as suggested (Now Fig. 6c, Suppl. fig 18 and Suppl. fig 19).